# Distinct structural groups of histone H3 and H4 residues have divergent effects on chronological lifespan in *Saccharomyces cerevisiae*

**Mzwanele Ngubo**[1¤], **Jessica Laura Reid**[2], **Hugh–George Patterton**[1,2]*

**1** Centre for Bioinformatics and Computational Biology, Stellenbosch University, Stellenbosch, South Africa,
**2** Department of Biochemistry, Stellenbosch University, Stellenbosch, South Africa

¤ Current address: Ottawa Hospital Research Institute, Regenerative Medicine Program, Ontario, Canada
* hpatterton@sun.ac.za

## Abstract

We have performed a comprehensive analysis of the involvement of histone H3 and H4 residues in the regulation of chronological lifespan in yeast and identify four structural groups in the nucleosome that influence lifespan. We also identify residues where substitution with an epigenetic mimic extends lifespan, providing evidence that a simple epigenetic switch, without possible additional background modifications, causes longevity. Residues where substitution result in the most pronounced lifespan extension are all on the exposed face of the nucleosome, with the exception of H3E50, which is present on the lateral surface, between two DNA gyres. Other residues that have a more modest effect on lifespan extension are concentrated at the extremities of the H3-H4 dimer, suggesting a role in stabilizing the dimer in its nucleosome frame. Residues that reduce lifespan are buried in the histone handshake motif, suggesting that these mutations destabilize the octamer structure. All residues exposed on the nucleosome disk face and that cause lifespan extension are known to interact with Sir3. We find that substitution of H4K16 and H4H18 cause Sir3 to redistribute from telomeres and silent mating loci to secondary positions, often enriched for Rap1, Abf1 or Reb1 binding sites, whereas H3E50 does not. The redistribution of Sir3 in the genome can be reproduced by an equilibrium model based on primary and secondary binding sites with different affinities for Sir3. The redistributed Sir3 cause transcriptional repression at most of the new loci, including of genes where null mutants were previously shown to extend chronological lifespan. The transcriptomic profiles of H4K16 and H4H18 mutant strains are very similar, and compatible with a DNA replication stress response. This is distinct from the transcriptomic profile of H3E50, which matches strong induction of oxidative phosphorylation. We propose that the different groups of residues are involved in binding to heterochromatin proteins, in destabilizing the association of the nucleosome DNA, disrupting binding of the H3-H4 dimer in the nucleosome, or disrupting the structural stability of the octamer, each category impacting on chronological lifespan by a different mechanism.

**Data Availability Statement:** The RNA-seq data was deposited in the NCBI GEO archive (accession number GSE141975). The Rap1 and Sir3 ChIP-seq data were deposited in the NCBI GEO archive

(accession numbers GSE141306 and GSE141317, respectively).

**Funding:** HGP 1U01HG007465 National Institutes of Health https://www.nih.gov/ The funding body did not contribute to the design of the study, collection, analysis, and interpretation of data, or to writing the manuscript.

**Competing interests:** The authors have declared that no competing interests exist.

## Introduction

In an attempt to understand the biochemical context of human diseases of aging such as cancer, diabetes, hypertension and cognitive decline, the regulation of lifespan has been studied as a controlled cellular process [1]. Many interconnected pathways have been implicated in the process of aging [1]. Although there is no clear understanding of the fundamental biochemical mechanism that allows prolonged cellular senescence and extended chronological lifespan, it is known that many of the signaling pathways involved in lifespan extension terminate at induced expression of stress related genes [2].

Calorific restriction is a common trigger in many organisms that feeds into the evolutionary conserved, carbon limitation regulatory pathways such as the TORC1-Sch9 or the Ras1-cAMP-PKA pathways [3, 4]. These two pathways converge on Rim15, which translocates to the nucleus when either the TORC1 or Ras1 pathways is inhibited, and facilitate binding of Msn2/Msn4 and Hsf1 to stress responsive and heat shock factor elements, respectively, inducing stress response genes [5]. The mitochondrial retrograde response represents an alternative pathway that signals mitochondrial stress to the nucleus. Here, the Rtg1-Rtg3 complex migrates to the nucleus and binds to R-boxes in conjunction with the SAGA-related SLIK1 acetyltransferase complex, inducing expression of the Rtg responsive genes, including CIT2, that catalyzes citrate synthesis in the glyoxylate cycle. All three pathways have an impact on chronological lifespan (CL) in yeast [6], but the full range of targets of these pathways are not known. Additional pathways that are involved in the regulation of lifespan include the mitochondrial Unfolded Protein Response [7] and induction of autophagy [8].

There is also increasing evidence that epigenetic changes are an integral part of the dynamics of aging [9], both as a result of aging, such as accumulation of H3K27me3 with time [10], and impacting on the regulation of aging [11, 12]. These epigenetic changes include histone modifications, histone modifying enzymes and transcription factor localization [12]. In addition, quiescent cells, a sub-population of yeast cells in chronologically aged yeast cultures, has been shown to be essential in determining the chromatin dynamics involved in longevity [13, 14]. The involvement of epigenetics was originally hinted at by the identification of the histone deacetylase, Sir2, in lifespan regulation [15]. Although this identification is now controversial [16], an association between SIRT6 and longevity in rodents was recently reported [17]. Furthermore, both high levels of H3K36 methylation [18] and the substitution of H4K16 was shown to increase replicative lifespan [19]. The yeast silencing complex formed by Sir2, Sir3 and Sir4 was also shown to be important in lifespan regulation [20], suggesting that heterochromatin is involved in this process [11]. Histone H3K4 methylation and H3T11 phosphorylation are the only histone modifications to date shown to affect chronological lifespan (CL) in yeast as opposed to representing a consequence of it [21, 22].

It was shown that transgenic mice, where a limited number of double strand breaks (DSBs) were made in the genome by induction of I-PpoI, displayed many features associated with old age at 10 months post treatment, including loss of visual acuity, muscle mass and neurological changes, when compared to an untreated control group [23]. Mouse embryonic fibroblasts derived from the same transgenic mouse line, displayed decreased H3K27ac, and H3K56ac, and increased H3K122ac levels in response to induced DSBs [24]. It was proposed that the repetitive repair of the DSBs causes a redistribution of histone modification marks, triggering the misregulation of a subset of genes that results in accelerated chronological aging [23, 24].

Here we focus on the reversed question, asking what the impact of histone modifications and epigenetic marks are on aging, as opposed to what epigenetic marks change as a result of aging. We identify categories of histone H3 and H4 residue substitution that impact on extension or reduction of CL, and demonstrate that substitutions with a profound impact on

lifespan extension may act through very different signalling pathways and transcriptomic programmes.

## Results

### Identifying histone H3 and H4 residues involved in the regulation of lifespan

We studied approximately 400 barcoded, non-lethal, synthetic histone H3 and H4 mutants in *Saccharomyces cerevisiae* [25], to gain an insight into the role of histone modifications in the regulation of CL. Each residue was systematically substituted with an alanine or a residue that mimicked the unmodified and modified state of an epigenetic switchable residue [25]. A culture that initially contained an equal level of each histone mutant strain was maintained in stationary phase for 55 d, quiescent cells isolated, and the level of each strain determined by independent quantitation of two DNA barcodes at progressive times. Quiescent cells were used to quantitate the level of healthy cells that were aging, as opposed to non-quiescent cells that were dying and had initiated apoptosis [26]. This "bar-seq" approach was previously shown to accurately quantitate individual strains in a mixture of barcoded yeast strains in culture [27]. We found that the level of some strains decreased at a lower rate, and others at a faster rate, compared to the population median or the parental WT (Fig 1a). The results of the quantitation by barcode sequencing are given in S1 Table. Although extended culturing of yeast in stationary phase increases acetic acid levels in the medium, this extended culturing is appropriate for studying CL, since it was shown that yeast faithfully reproduced responses and induction of pathways associated with chronological aging observed in other model organisms that were not exposed to elevated levels of acetic acid [28].

### Residues with the largest impact on chronological lifespan extension are exposed on the nucleosome disk face

The 10 individual residues that caused the most pronounced lifespan extension (red residues in Fig 1a and 1c) are all located either in the H4 tail or exposed on the solvent accessible face of the nucleosome, within the sectors previously identified as the Swi/Snf independent (Sin) or the Loss or rDNA silencing (Lrs) sectors at SHL±0.5 and ±2.5 [29]. H4K16 Q and R substitutions were previously shown to prolong replicative aging [19]. We have identified H4K16 as a residue also implicated in chronological lifespan extension (CLE) (S1 Table). Interestingly, any mutation of H4K16 (to Q, R, or A) cause an increase in CL (S1 Table), suggesting that a lysine residue is specifically required at position 16 for normal CL. This suggestion is supported by the observation that both the H4Δ9–16 and H4Δ12–16 tail deletions extend lifespan (Fig 1i and S1 Table). Interestingly, the H4 tail deletions Δ1–12 extended CL, whereas the short N-terminal deletion Δ1–4 reduced lifespan, demonstrating that the combined loss of H4 residues 5–12 were sufficient to abrogate the negative effect of loss of residues 1–4 (Fig 1i and S1 Table). This is supported by the extended lifespan observed for the Δ5–12 H4 deletion (Fig 1i and S1 Table).

The solvent exposure of the top 10 individual residues where mutation prolong lifespan suggests that these residues are involved in binding to a protein on the surface of the nucleosome disk. It is known that H4K16, H4H18, H4L22, H4N25, and H3T80, identified in the strains with extended lifespan, all interact with the Sir3 BAH domain [30, 31]. Sir3 is involved in establishing transcriptionally repressive heterochromatin at the telomeres and silent mating type loci. The only residue in this solvent exposed group that is not known to interact with the BAH domain is H3E50. H3E50 is exposed on the lateral surface of the nucleosome core

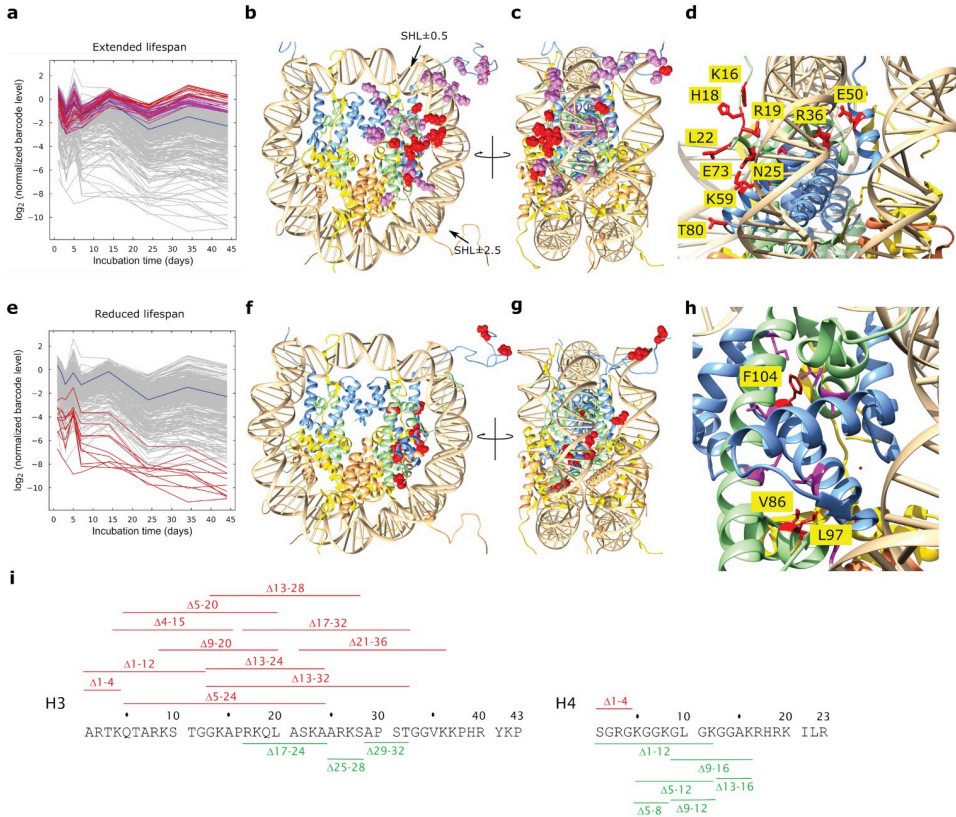

**Fig 1. Chronological lifespan of histone mutants.** The barcode level of each of the approximately 400 tagged histone mutant strains was determined by sequencing in the quiescent pool at different times in stationary phase, and expressed as $\log_2$ relative to the level at the start of stationary phase (6d), and adjusted to the population median. The $\log_2$ ratios of the top 10 individual (red) and top 10% (violet) of strains with most extended lifespan (**a**) and bottom 10 individual strains (red) with most reduced lifespan (**e**) are shown at progressive times of incubation. The $\log_2$ ratio of the median strain is indicated (blue). The residues implicated in the top 10 individual and top 10% of strains that exhibit extended lifespans are shown in red and violet sphere conformation in the structure of the nucleosome (1KX5), respectively (**b**), and the bottom 10 individual strains with the most reduced lifespans are shown in red sphere conformation (**f**). The positions of superhelix locations (SHL) ±0.5 and ±2.5 aligned with the Sin and Lrs sectors [29] are indicated by arrows (**b**). The structures in **b** and **f** are shown rotated by 90° clockwise viewed from the top (**c, g**). The 10 individual residues associated with most extended (**d**) and most reduced (**h**) lifespan are shown in stick conformation with individual residues identified. The deletions of sections of the N-terminal tails of H3 and H4 that are associated with extended (green) or reduced (red) CL are indicated (**i**). Individual residues atoms of the mutated residue in the top 10 most viable (**b**) and 10 least viable (**e**) strains in the quiescent fraction are shown as red spheres in the crystal structure of the nucleosome (PDB accession 1KX5). A close-up view of the nucleosome shows the location of the mutated residues, represented in red stick structures, in the most (**c**) and least viable (**f**) strains.

between two DNA gyres (Fig 1c and 1d), suggesting that H3E50 may influence the regulation of yeast lifespan through a different, Sir3-independent pathway. H3E50 makes an H-bond to H4R39 in α1 of H4, which could contribute to placing the N-terminal tail of H3 in the correct radial position in the nucleosome (S1a Fig).

## The top 10% of residues that cause moderate chronological lifespan extension are concentrated at the H3-H4 dimer extremities

Strikingly, the top tenth percentile of residues associated with prolonged CL, excluding those on the solvent exposed face of the nucleosome and H3E50, are concentrated at the extremities

of the histone fold domain (violet residues in Fig 1a). Residues in this category appear to be concentrated at the H3 α3 C-terminus, the C-terminal region of α2 and L2 of H3, the N- and C-terminal regions of α1 and L1 of H3, the N-terminal region of H4 α2, bracketing the H4 L2 at the C-terminus of α2 and N-terminus of α3, and in the region where H4 α1 passes over H3 αN (Fig 1b and 1c). The interactions of these residues specifically exclude the extensive interactions between dimer partners, and seem enriched for contacts that stabilize the conformation of the histone fold domain and the H3-H4 dimer in its frame within the nucleosome structure (see Supplementary S1 Note). It is not immediately clear what the impact of mutations at these positions are on dynamic nucleosome structure.

## Residues implicated in a reduced CL are buried, and may destabilise the octamer

The positions of the mutated residues in the ten strains with most reduced lifespans (Fig 1e) in the quiescent pool are shown in Fig 1f and 1g. Most H3 tail deletions, including the short N-terminal deletion Δ1–4, resulted in reduced lifespan, showing that an intact H3 tail is required for normal lifespan (Fig 1i and S1 Table). In contrast, short internal H3 tail deletions Δ17–24, Δ24–28 and Δ29–32 extends lifespan (Fig 1i and S1 Table). The first two of these short deletions bracket the residues K23 and K27, both in the top 10% of residues that extend lifespan when mutated (Fig 1i and S1 Table). Notably, substitution of four histone H3 tail lysine residues with arginine (H3K4, 9, 14, 18R) reduce lifespan. Either this suggests that mimicking a constitutively deacetylated state for these lysine residues decreased lifespan, or that a modifiable lysine is required for normal lifespan. The latter seems likely, since the H3K18Q mutant, mimicking a constitutive acetylated state, also exhibits a shortened lifespan (S1 Table). In addition, deletion of Set1, which abrogates methylation of H3K4, decreased CL, and, conversely, deletion of the H3K4me3 demethylase, Jhd2, increased CL [21], arguing that a modifiable H3K4 residue is necessary for a normal CL.

The location of the residues where mutations decreased lifespan is summarized in Fig 1f and 1g. It is striking that, unlike the solvent exposed residues (Fig 1b and 1c) or residues implicated in stabilizing each H3-H4 dimer in a structural frame (Fig 1b and 1c), all residues implicated in a shortened lifespan, except for the H3 N-terminal tail mutations, are buried within the octamer structure, and occur between interacting sections of the histone fold domains (Fig 1f and 1g). We speculate that the mutations that cause a shortened lifespan contribute to the structural destabilization of the histone octamer and disruption of nucleosomes and chromatin structures (Supplementary S1 Note). It was previously shown that reducing histone H3 and H4 levels, and presumably nucleosome density, resulted in a reduced CL [32].

We verified the results of the barcode approach by confirming the survival of select strains implicated in lifespan regulation in biological replicates (Fig 2). The reproducibility of the observed lifespans also shows that the result of the bar-seq quantitation does not simply reflect the population variance of a random process.

## The H4K16 and H4H18 mutants cause re-distribution of Sir3 in the genome

The solvent exposed residues H4K16, H4H18, H4L22, H4N25 and H3T80 all interact with the BAH domain of Sir3 in the crystal [30], and substitution of these residues cause CLE. These substitutions are expected to disrupt the binding of Sir3 on the side of the nucleosome [30, 31]. To investigate the effect of mutation of these residues on the genomic distribution of Sir3, we performed a ChIP-seq analysis of H4K16Q, H4H18A and H3E50A mutants. These mutations were selected to include one residue previously implicated in lifespan extension, H4K16, albeit

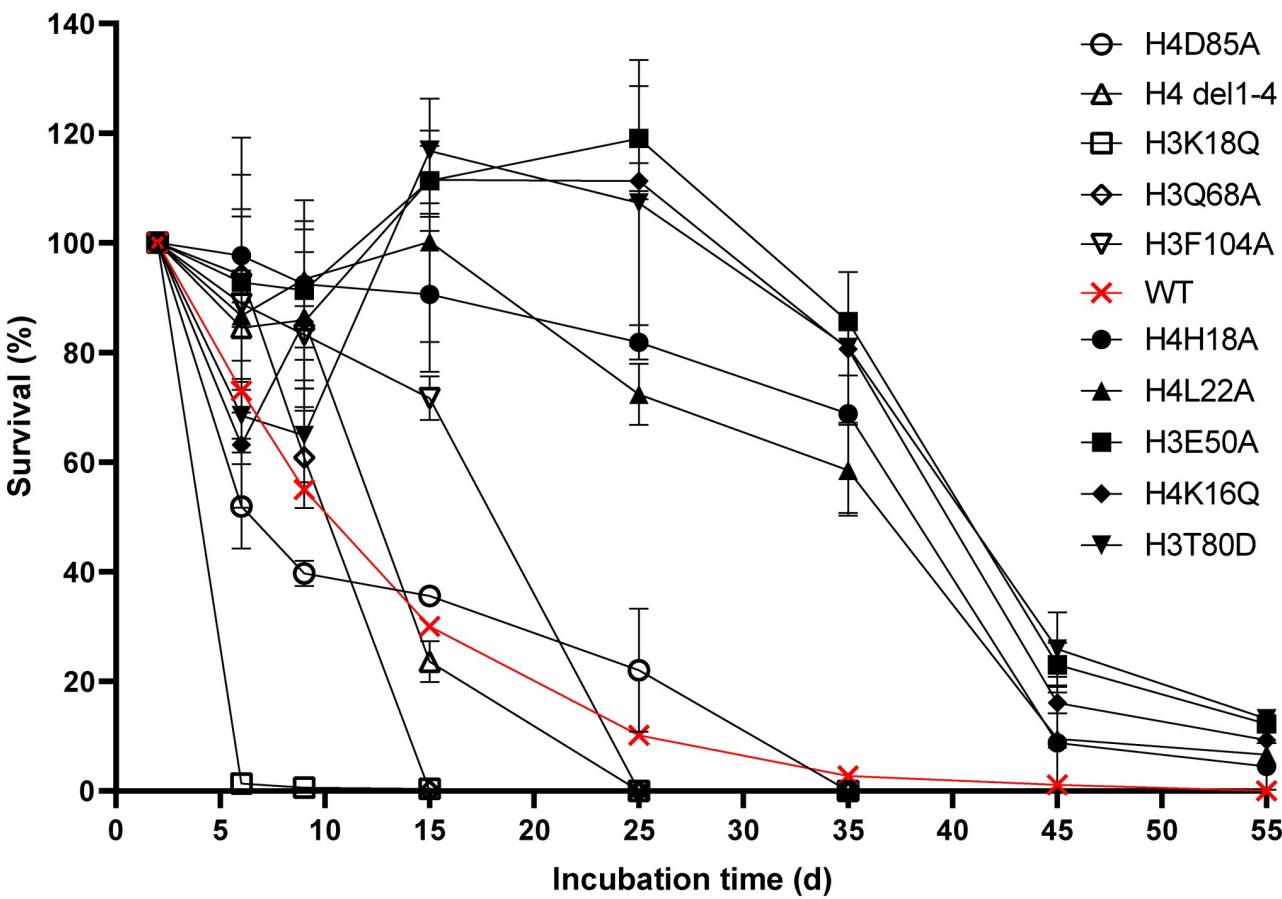

**Fig 2. Growth curve of selected strains with extended and truncated lifespans.** Strains with extended or shortened lifespans were selected and cultured individually for up to 55 days. The cell density of each strain was determined at different times, and the percentage cells remaining relative to the starting culture is shown at the different culture times. The strain with the *WT* histones H3 and H4 are indicated by the red plot. Experiments were performed in duplicate. Error bars show the standard deviation (SD).

replicative lifespan [19], a residue newly identified to cause CLE, H4H18, and a residue similarly shown to cause CLE, but unlikely to be involved in direct Sir3 binding, H3E50.

After incubating cultures until stationary phase for 6 days, the Sir3 occupancy at HMLα and HMR**a** were found to be reduced by at least two-fold in the H4K16Q and H4H18A mutants, and to approximately 80% of the WT level at HMLα in the H3E50A mutant strain (Fig 3a and 3c). Indeed, in the exponential and stationary phase WT, Sir3 occupancy at HM loci is enriched than in the H4K16 and H4H18 mutant strains (Fig 3b and 3d), suggesting that Sir3 translocates from the telomeres and mating type loci in stationary phase in the mutant strains. In a WT cell Sir3 was enriched at the telomere X element and the borders of the Y' element, as previously reported [33]. Sir3 binding was reduced at the X elements in the H4K16Q, H4H18A and H3E50A strains, but remained at levels comparable to the WT at the Y' element (Fig 3e). When looking at Sir3 distribution throughout the whole genome, it is seen that the levels of Sir3 at the terminal 20 kb of chromosomes, normalised to the genome, are significantly lower (*t*-test, $p < 0.05$) in the H4K16Q and H4H18A mutants compared to the WT (Fig 4). There is no significant difference in Sir3 telomeric levels between the WT and the H3E50A mutant, but there is a significant difference between the levels between the H4K16Q or H4H18A mutant and the H3E50A mutant (*t*-test, $p < 0.05$). A similar pattern of differences is

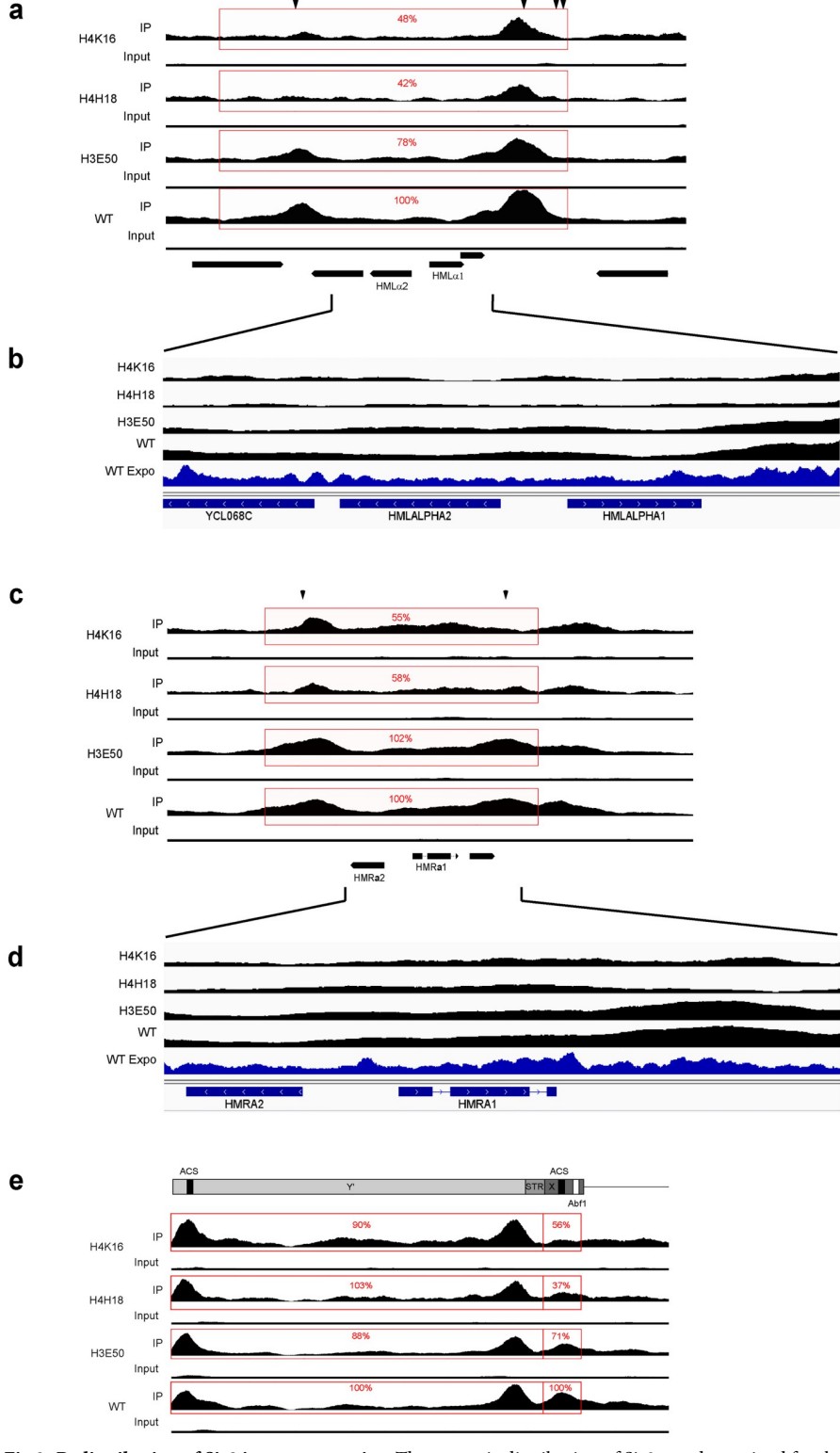

**Fig 3. Redistribution of Sir3 in mutant strains.** The genomic distribution of Sir3 was determined for the H4K16Q, H4H18A, H3E50A and WT strains. The immuno-precipitated and input signal, normalised to a ratio of 1, is shown for each strain at the HMLα (**a**). (**b**) HMLα zoomed-in IGV tracks of (**a**) compared with publicly available exponential phase Sir3 ChIP-seq WT (WT Expo) in blue track from [36]. (**c**) Sir 3 distribution at HMRa, (**d**) HMRa zoomed-in IGV tracks of (**c**) compared with publicly available exponential phase Sir3 ChIP-seq WT and left telomere of

chromosome IX (**e**). The arrows indicate the positions of autonomously replicating consensus sequences (ACSs). The percentage of the pull-down signal relative to the WT strain is shown for each track (representative for biological replicates, n = 2). The positions of the MATα1, MATα2, MATa1 and MATa2 genes are indicated. The line diagram at the top of panel (**e**) shows the location of the X and Y' telomeric elements, the ACSs, Abf1 binding site and sub-telomeric repeat sequences (STR).

seen between the Sir3 levels at the chromosome cores, excluding the terminal 20 kb (Fig 4). This demonstrates that Sir3 is redistributed from the telomeres to the chromosome cores in the H4K16Q and H4H18A mutants. This may reflect a migration from a compromised nucleosome binding site to secondary binding sites in the genome with binding affinities that are now competitive with the mutant nucleosome. No significant change in the telomeric distribution of Rap1, which recruits Sir3 to the telomeres to form silent heterochromatin [33], is seen when comparing the H4H18A mutant to the WT strain, underlining the fact that Rap1 binds to DNA [34] whereas Sir3 associates with nucleosomes (Fig 5). It was previously shown that Rap1 redistributed from the telomeres to internal sites in a Δ*tlc1* strain with shortened

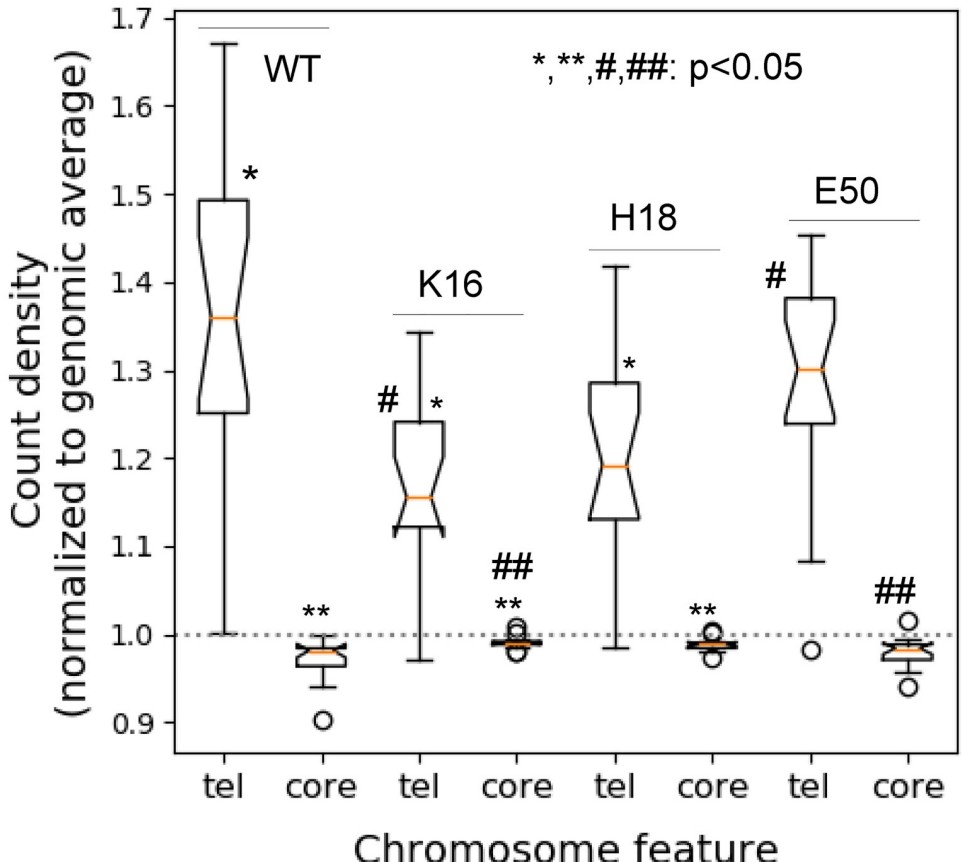

**Fig 4. Distribution of Sir3 between telomeres and chromosome cores.** The level of Sir 3 associated with the 20kb chromosome termini and the internal core regions excluding the 20 kb termini were determined from the ChIP-seq signal for each of the 16 chromosomes, and normalised to the genome average. The values are shown for the WT and H4K16Q, H4H18A and H3E50A mutant strains. The box plots represent the inter-quartile range (IQR), the notch represents the significance at p<0.05, the orange line shows the median, and the whiskers is shown at 1.5× the IQR. Outliers are shown as individual data points. WT and mutants strains and mutant-mutant strain pairs that are significantly different (*t*-test; p<0.05) in the telomeric and core regions are indicated by the * and # and by ** and ## symbol pairs, respectively.

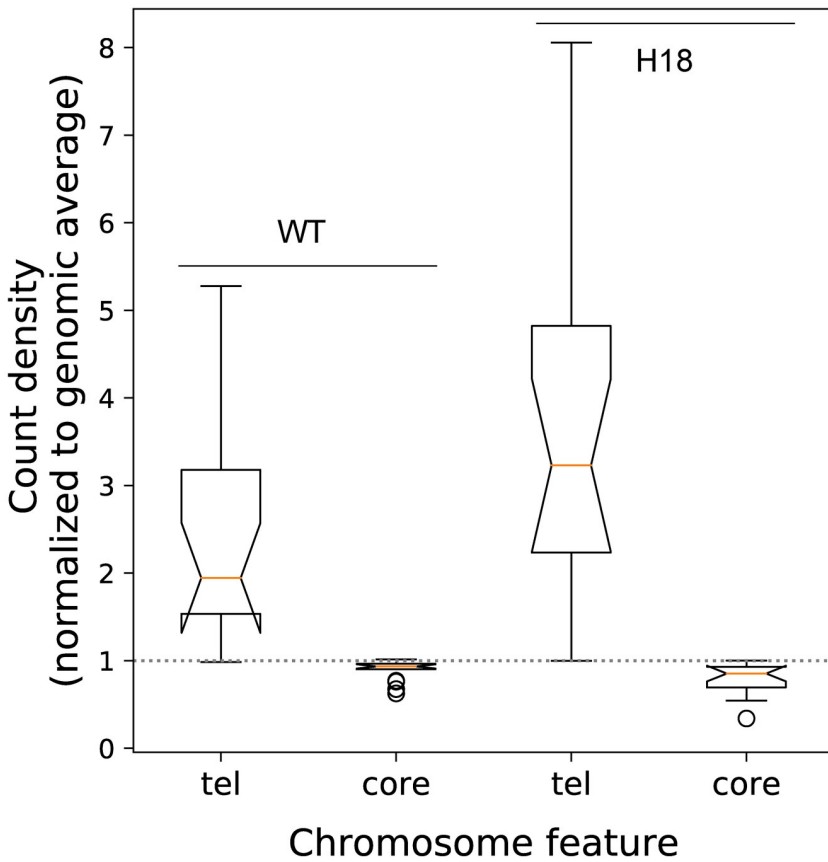

**Fig 5. Rap1 binding in the genome.** The level of Rap1 bound to the 20kb chromosome termini (tel) and the internal core regions excluding the termini (core) were determined from the ChIP-seq signal for each of the 16 chromosomes, and normalised to the genome average. The values are shown for the WT and H4H18A mutant strains. The box plots represent the inter-quartile range (IQR), the notch represents the significance at p<0.05, the orange line shows the median, and the whiskers is shown at 1.5× the IQR. Outliers are shown as individual data points.

telomeres, which was proposed to represent an aged senescent cellular state [35]. However, this relocation was likely due to decreased Rap1 binding sequence and not simulated age, since we saw no evidence of such a reorganization with chronological aging over a 14 d period (S2 Fig).

### Re-distributed Sir3 is associated with transcriptional repression at the new loci

Sir3 peaks were identified in the ChIP-seq data of the mutant strains. To assess whether the redistributed Sir3 repressed transcription in the newly occupied regions, we analysed the differential expression of genes in the mutant strains relative to the WT when the transcribed gene, including a 500 bp upstream region, overlapped with the identified new Sir3 peak. In the H4K16Q mutant, 33 genes overlapped with Sir3 peaks, of which 22 genes were repressed compared to the WT. Given that 48% of the quantitated genes in the complete RNA-seq data set were down-regulated, a binomial probability distribution function of $\binom{33}{22} \times 0.48^{22} \times 0.52^{11} = 0.01$ is obtained. Thus, there is only a 1% chance that random selection of 33 genes from this RNA-seq data set will include 22 down-regulated genes, strongly suggesting that the redistributed Sir3 indeed represses gene expression at most of the newly occupied loci.

There is no unique transcription factor binding site associated with all re-distributed Sir3 peaks. Importantly, Sir3 peaks were not always associated with an ARS consensus sequence, found in the E elements of the HM cassettes. We calculated the significance of the presence of factor binding sites in Sir3 peaks given the number of such factor sites in the genome, and assuming a Poisson probability distribution (S2 Table). It seems that Sir3 distributes to a number of sites, recruited by a combination of factors including Abf1, Rap1 and other factors (S2 Table). It therefore appears likely that Sir3 redistributes to a number of diverse secondary binding sites in the genome in the H4K16 and H4H18 mutant strains.

Considering the genes associated with Sir3 peaks in the mutant strains, we found that the YIL055C gene is common to all three extended CL mutant strains (S3 Fig and S3 Table). YIL055C encodes a protein of unknown function that is associated with the mitochondrion and interacts genetically with the histone deacetylases Hda1 and Hos1, subunit 8 of ubiquinol cytochrome-c reductase (Complex III), the β-subunit of the Sec61-Sss1-Sbh1 ER translocation complex, actin and Cdc13, the telomere repeat binding protein involved in the regulation of telomere replication. A Δyil055c strain is, however, not associated with CLE [37, 38].

The redistributed Sir3 peaks of the H4K16Q and H4H18A mutant strains commonly cover MAM3, encoding a protein associated with the ER membrane that is required for mitochondrial morphology, and SND1, encoding a protein involved in alternative ER targeting. Again, neither Δmam3 nor Δsnd1 strains are associated with CLE [37, 38] (Fig 6a–6d).

It was previously shown that the propagation and termination of repressive heterochromatin domains occur with a degree of randomness that is the basis for position effect variegation [39]. It is thus expected that the exact positions of the new loci associated with Sir3 will differ between cells and between strain, which could allow different genes that have an impact on the regulation of lifespan to be repressed in the otherwise identical H4K16 and H4H18 mutant strains. The GDH1 glutamate dehydrogenase gene is repressed in the H4K16 mutant strain, and is associated with a CLE phenotype in a null strain [37]. In the H4H18 mutant strain the AGP1, HTZ1 and SHR5 genes, encoding a glutamine transporter, the histone H2A.Z, and a palmitoyltransferase that suppresses Ras1 function, are associated with new Sir3 loci, and were all shown to be associated with CLE phenotypes in null mutants [37, 38]. The redistributed Sir3 is thus associated with genes that were previously shown to confer extended CLs in null mutants, and could provide a simple, causative link between Sir3 redistribution and lifespan expansion.

## The H4K16Q and H4H18A mutants have similar transcription profiles that differ from that of H3E50A

Although it is possible that the repression of one or a few genes by redistributed Sir3 may impact on the expression of a central and important component in a pathway involved in the regulation of CL, and single genes identified above may contribute to lifespan extension, we note that the transcriptomic profile of the CLE strains differ from that of the WT strain at hundreds of genes. It is thus likely that the histone mutations also have gene regulatory effects other than Sir3 binding, or that the redistribution of Sir3 in the genome have indirect effects that impact on many additional genes, some of which are involved in CLE. We therefore analysed the RNA-seq data of three CLE strains to attempt to identify functional GO categories or pathways that presented an additional possible mechanistic basis for the observed extended CL.

Gene expression is significantly down-regulated in stationary phase [40]. Although the expression or repression of specific genes in stationary phase may support CLE, we were interested in the transcription programme that preceded stationary phase, and that may prepare a

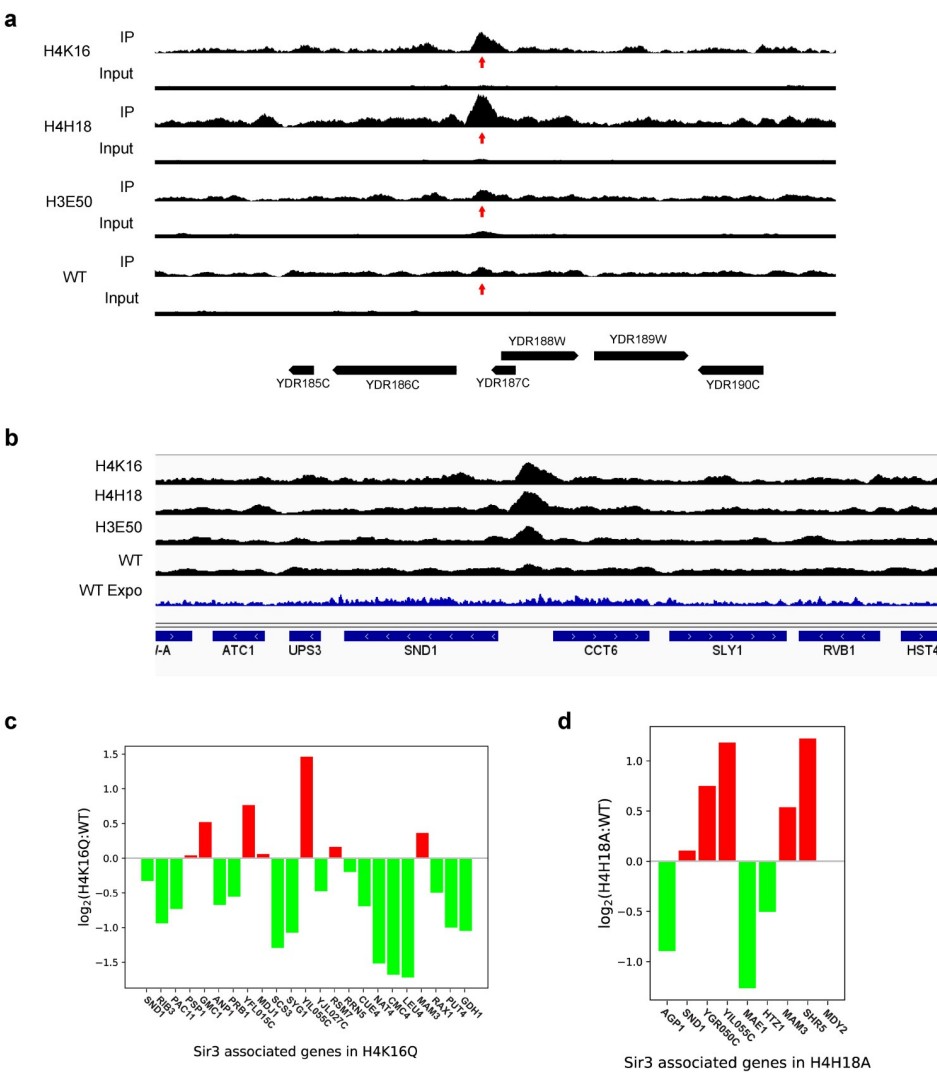

**Fig 6. Binding of Sir3 to secondary loci in the genome.** (**a**) The normalised level of bound Sir3 in a region of the right arm of chromosome IV is shown for the H4K16Q, H4H18A, H3E50A and WT strains. The arrow identifies a Sir3 peak present upstream of the YDR186C (SND1) gene in the H4K16Q and H4H18A strains, but not in the H3E50A or WT strains. (**b**) IGV Sir3 ChIP-seq tracks for the H4K16Q, H4H18A, H3E50A and WT strains at day 6, and the blue track of WT Sir3 ChIP-seq at exponential phase (Expo) from [36]. Bar graphs showing average log2 fold change for the genes associated with secondary Sir3 peaks in H4K16Q (**c**) and H4H18A strains (**d**). Wald test estimated the standard error of log2 fold change using two biological replicates.

biochemical state that allowed subsequent extended CL in stationary phase. For this reason, we performed RNA-seq analyses on cultures in late log phase.

The H4K16 and H4H18, H4K16 and H3E50 and H4H18 and H3E50 RNA-seq datasets have Spearman rank correlation coefficients of 0.7, -0.04 and -0.2 respectively, showing that H4K16 and H4H18 are most closely related, and that the H3E50 profile has little correlation to either of the H4 mutant strain sets.

An analysis of the GO terms for biological processes enriched for the genes that are induced in the H4K16Q strain (adjusted $p<0.05$) compared to the WT show a significant enrichment for biosynthetic processes, cytoplasmic translation, ribosome biogenesis and RNA processing (S4 Table). The largest number of induced genes further map to pathways involved in the

biosynthesis of secondary metabolites and amino acids, ribosome biogenesis, RNA transport and purine metabolism (S5 Table). Genes that are down regulated in the H4K16Q strain (adjusted p<0.05) are associated with DNA integration, recombination and biosynthetic processes, as well as mitochondrial respiratory chain complex assembly (S6 Table). Down regulated pathways include metabolic, cell cycle, MAPK signalling, autophagy and carbon metabolism pathways (S7 Table). It thus appears that in late log the cell is actively synthesizing proteins, but shutting down oxidative phosphorylation and DNA synthesis. A very similar pattern of regulated genes and enriched GO functional categories and pathways was observed for the H4H18A strain (S4–S7 Tables).

In contrast, the H3E50A mutant shows increased expression of genes associated with oxidation-reduction processes, generation of precursor metabolites, energy derivation by oxidation of organic compounds, tricarboxylic acid cycle and the electron transport chain (S4 Table). Enriched pathways include carbon metabolism, tricarboxylic acid metabolism, respiratory electron transport chain, oxidative phosphorylation and autophagy (S5 Table). Down regulated genes include the GO functional categories of ribosome biogenesis, cytoplasmic translation, RNA processing (S6 Table), ribosome biogenesis, cell cycle, and purine metabolism pathways (S7 Table).

The transcriptomic profile of the H3E50A mutant strain is essentially the inverse of the H4K16Q and H4H18A strains. Protein synthesis is down regulated, and the tricarboxylic acid cycle and the oxidative phosphorylation is activated. It thus appears that the H3E50A strain is actively synthesizing ATP by oxidative phosphorylation in late log phase. MSN4, CIT2, SOD1 and SOD2 are all significantly induced in the H3E50A strain, suggesting that the mitochondrial retrograde response is active, and that the yeast cell is responding to oxidative stress and inducing Msn4 responsive stress genes. It is surprizing that the mutation of a single residue that stabilises the position of the N-terminal tail of histone H3 in the nucleosome induces a mitochondrial retrograde response or related transcriptomic effect. The H4K16Q and H4H18A strains display elevated MSN2/4 expression levels, but decreased CIT2, SOD1 and SOD2 levels, suggesting the absence of the retrograde response, and the presence of a general stress response in these mutant strains.

## Discussion

We have identified four groups of H3 and H4 residues situated in different regions of the nucleosome where substitution have an impact on CL. The location of each group provides a hint of the possible mechanistic route by which it influences CL. The first group consists of residues exposed to the solvent on the face disk of the nucleosome and present in the N-terminal tails. The residues on the face disk are involved in binding to the Sir3 heterochromatin protein and cause a pronounced extension of CL. The operative substitution of these residues do not necessarily represent valid epigenetic switches. H4K16, for example, confers CLE when substituted with either R, Q or A. It seems that an unmodified K residue stabilises binding to Sir3 by forming a hydrogen bond between the ε–amino group and the S67 hydroxyl in the BAH domain of Sir3 [30], an association that is necessary for a normal CL. Any alteration disrupts this H-bond, irrespective of whether it is an epigenetic mimic or not, and extends CL. Another example is H4H18, a residue that was also shown to interact with Sir3 by formation of an H-bond between the imidazole imino group and E95 in the BAH domain [30]. H4H18 extents CL when mutated to an A, which does not represent a valid epigenetic state switch.

H3T11, on the other hand, present in the N-terminal tail of H3, extends CL when substituted with D, a mutation that mimics constitutive phosphorylation. The WT residue and the H3T11A mutant exhibit a CL similar to that of the population average. This category of residue

is of acute interest, since it represents an epigenetic mark where transition between different epigenetic states modulate CL.

The second group of residues in the nucleosome is defined by the single member, H3E50. This residue, like the members of the first group, causes prolonged CL when mutated, but is not involved in binding to Sir3. The WT residue makes an H-bond to an amino group in H4R39, and contributes to setting the exit position of the H3 N-terminal tail in the nucleosome. When substituted in the H3E50A mutant, CL is extended. It was previously shown that the H3E50A mutant has a double strand break checkpoint defect [41], suggesting an alternative mechanistic pathway by which a histone residue can effect CLE.

The third group of residues are clustered at the extremities of the histone folds of H3 and H4, at L1 and L2, and are likely to be involved in stabilizing the conformations of the α1 and α3 helices relative to α2, as well as binding to the DNA duplex. These residues may be involved in stabilizing the H3-H4 dimer in its structural frame in the nucleosome. It is possible that substitutions at the L1 and L2 positions influence H3-H4 dimer binding in the nucleosome and are likely to have a pleiotropic effect and an impact on diverse functionalities of chromatin. This category of substitution is likely to confer CLE through several pathways since it may involve disparate functions of the genome. Residues in group three typically extend CL to a lesser degree than residues in groups one and two.

The fourth category is composed of residues buried within the octamer and are likely involved in stabilizing the histone octamer itself. Residues in the fourth category are exclusively associated with a shortened CL when substituted. We postulate that these mutations disrupt the structural stability of the nucleosome, a fundamental structural unit of chromatin, and may accelerate apoptosis. It was previously shown that loss of histone H3 and H4 reduced CL [32].

The Workman group showed that H3T11 was phosphorylated by both Sch9 and Cka1, and reported that Δsch9 and Δcka1 as well as H3T11A prolonged CL [20]. Although this seems opposite to our result where the phosphorylated mimic H3T11D showed CLE, we note that the Δsch9 mutant strain will facilitate activated expression of Msn2/4 stress response genes associated with CLE [42], and it is not clear if the CLE observed was due to absence of H3T11 phosphorylation or induction of stress response genes. A phosphorylated mimic H3T11D was not tested [20].

A SPELL analysis [43] showed that both the H4K16 and H4H18 mutant gene expression profiles were most closely related to that of yeast strains under conditions of carbon and nitrogen stress or DNA replication stress. Importantly, neither the H4K16 nor H4H18 mutant strain exhibited a gene expression profile similar to a sir3Δ strain. The H3E50 transcriptomic profile matches most closely sets related to stationary phase entry, carbon utilization, diauxic shift, fermentation and respiration.

A search of the HistoneHits database [44] of the classic phenotypes associated with histone mutants showed that H3E50A was associated with a modest decrease in the DNA damage phenotype. H4K16A and H4K16Q showed decrease in telomeric silencing and mating efficiency, and H4K16A, H4K16Q and H4K16R showed HM cassette derepression. H4H18A and H4H18Q displayed defects in HM and telomeric silencing. The majority of H4 tail deletions between 1 and 28 exhibited HM cassette silencing defects. We note that although chronological aging is associated with HM derepression, overexpression of MATalpha1 gene in haploid MATa strain is associated with decreased chronological aging [45].

We propose that the redistribution of Sir3 to alternative sites in the genome is caused by a change in the affinity of the binding site on the face of the nucleosome disk for the Sir3 BAH domain, due to the substitution of an interacting histone residue. In *S. cerevisiae*, heterochromatin is typically initiated by the recruitment of a heterochromatin protein such as Sir1 to a DNA-bound initiation factor such as Orc1 [46]. The heterochromatin domain is propagated by the binding of a modifying enzyme such as the Sir2 deacetylase to the heterochromatin

initiation core, deacetylation of H4K16ac of the adjacent nucleosome, binding of the hetero-chromatin protein complex Sir3/Sir4 to the modified nucleosome, and re-recruitment of Sir2 by the newly deposited Sir3/Sir4. The continuous, repetitive modification of each adjacent nucleosome and subsequent binding of Sir3/Sir4 leads to the progressive extension of the heterochromatin domain along the DNA [47, 48].

Both the initiation factor as well as the nucleosome contribute to define a binding affinity, and this initial affinity will influence successive binding affinities in the propagated hetero-chromatin domain due to contact between the neighbouring Sir3/Sir4 sub-units [49]. This propagation continues with a defined probability with every consecutive, interacting Sir3/Sir4 sub-unit, or until firm termination by an insulator, TFIIIC bound to a pol III gene A-box [49] or a euchromatic domain defined by an epigenetic modification such as H3K79me3 [50] or the enrichment of histone variant Htz1 [51].

We did not detect any transcription factor binding site that was invariably associated with the redistributed Sir3 domains in the mutant strains. A range of sites were linked to the new loci, suggesting that Sir3 is recruited to different regions of the genome by a number of different or even a mixture of initiator factors such as Abf1 and Rap1, known to bind to Sir3 [52]. When comparing the Sir3 alternative sites in the mutant strains and the secondary binding sites induced by Sir3 overexpression [53, 54], we observe minimal overlap, suggesting that the recruitment of Sir3 to the alternative sites that cause lifespan extension does not resemble the same recruitment patterns as in cells with ectopic Sir3 expression. Therefore, our proposed model predicts systematic redistribution of Sir3 to regions that are key for CLE, contrary to increased concentrations of Sir3. These secondary binding sites will also have an affinity defined by a combination of the recruiting factor and the binding site on the nucleosome face.

We considered the validly of such a redistribution model by considering the equilibrium binding of Sir3 to two classes of binding sites (Fig 7). The binding of Sir3 to a high affinity, primary site, such as an HM locus, can be described by the equilibrium equation

$$[P] + [B_1] \rightleftharpoons [PB_1] \tag{1}$$

where $[P]$ and $[B_1]$ represents the concentrations of Sir3 and the nucleosome binding site, respectively, and $[PB_1]$ is Sir3 bound to the binding site. The association constant $K_{a1}$ is defined by

$$K_{a1} = \frac{[PB_1]}{[P][B_1]} \tag{2}$$

Similarly, the binding to a low affinity, secondary binding site can be represented by

$$[P] + [B_2] \rightleftharpoons [PB_2] \tag{3}$$

where $[B_2]$ represent the concentration of the secondary binding site and $[PB_2]$ is bound Sir3.

The fractional binding $f_{PB_1}$ of Sir3 associating with the primary site $B_1$, assuming that $[P]$ is limiting, thus $[P] \ll [B_1]$, is given by

$$f_{PB_1} = \frac{[PB_1]}{[P] + [PB_1] + [PB_2]} \tag{4}$$

$$= \frac{K_{a1} \cdot [P] \cdot [B_1]}{[P] + K_{a1}[P] \cdot [B_1] + K_{a2} \cdot [P] \cdot [B_2]} \tag{5}$$

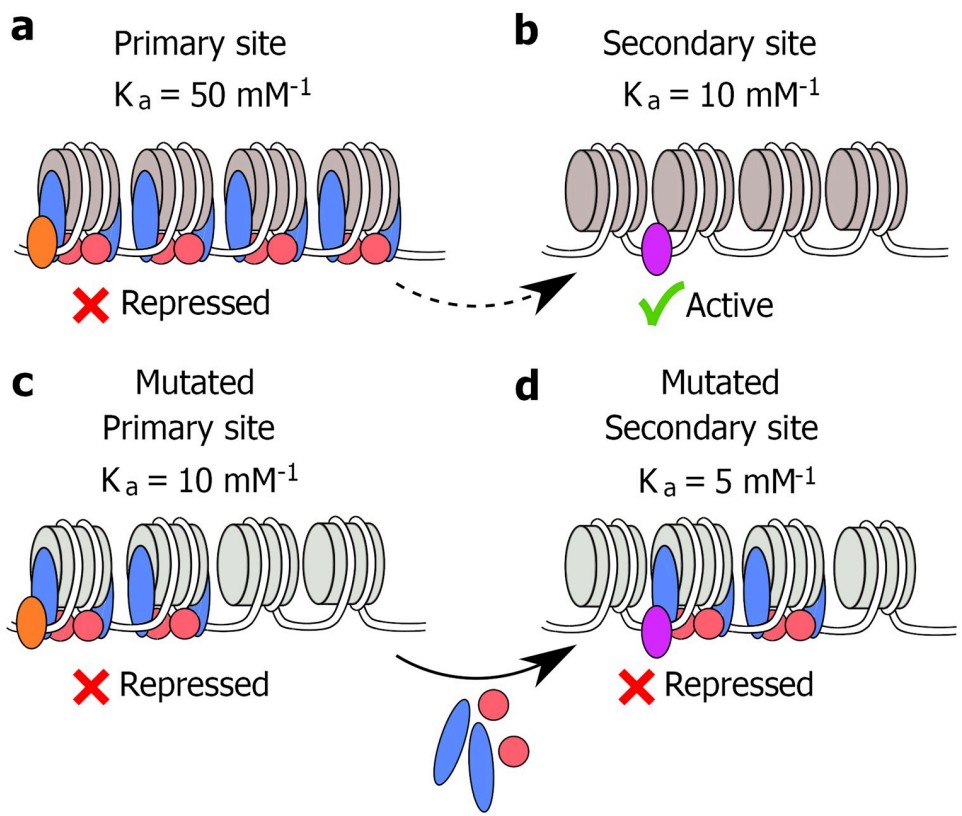

**Fig 7. A model for the redistribution of Sir3 in the genome of some histone mutant strains.** The Sir3/Sir4 complex is recruited to a high-affinity binding site by a DNA bound factor to form a heterochromatin domain that represses the covered gene (**a**). The affinity of Sir3/Sir4 for the secondary binding site is significantly less, and little or no Sir3/Sir4 complex binds to this site, allowing the associated gene to remain transcriptionally active (**b**). The mutation of a residue on the nucleosome face that interacts with Sir3 decreases the affinity for the primary binding site significantly (**c**). Although the affinity of the secondary binding sites also decreases, the presence of a different DNA bound factor at this site may modulate this decrease (**d**). The comparable affinities of the mutated primary and secondary binding sites allow Sir3/Sir4 to migrate from primary to secondary binding sites, causing repression of genes associated with the secondary binding sites (**d**).

Fractional binding of Sir3 to the secondary site $B_2$ is similarly given by

$$f_{PB_2} = \frac{[PB_2]}{[P] + [PB_1] + [PB_2]} \tag{6}$$

$$= \frac{K_{a2} \cdot [P] \cdot [B_2]}{[P] + K_{a1} \cdot [P] \cdot [B_1] + K_{a2} \cdot [P] \cdot [B_2]} \tag{7}$$

Assuming that $[B_1] \approx [B_2] = [B]$, which appears likely, judged by the comparable number of Sir3 peaks observed in the WT and the mutant strains, gives

$$f_{PB_1} = \frac{K_{a1} \cdot [B]}{1 + [B] \cdot (K_{a1} + K_{a2})} \approx \frac{K_{a1}}{K_{a1} + K_{a2}} \text{ if } B \gg 1 \tag{8}$$

$$f_{PB_2} = \frac{K_{a2} \cdot [B]}{1 + [B] \cdot (K_{a1} + K_{a2})} \approx \frac{K_{a2}}{K_{a1} + K_{a2}} \text{ if } B \gg 1 \tag{9}$$

Setting $K_{a1}$ and $K_{a2}$ of the high and low affinity binding sites equal to 50 μM$^{-1}$ and 10 μM$^{-1}$, implies a $f_{PB_1} = \frac{5}{6}$ and $f_{PB_2} = \frac{1}{6}$. The high affinity binding sites are thus bound at a 5-fold higher level than the low affinity binding sites. If the $K_{a1}$ in the mutant strain is decreased by 5-fold from 50 to 10 μM$^{-1}$ due to the disruption of the Sir3 binding surface on the nucleosome face, and $K_{a2}$, representing a different initiator factor in combination with a nucleosome face, is reduced 2-fold from 10 to 5 μM$^{-1}$, $f_{PB_1} = \frac{4}{6}$ and $f_{PB_1} = \frac{2}{6}$. Thus, a 20% reduction in binding to the primary site and a 100% increase in binding to a secondary site will be evident.

We propose that we observe a related binding scenario in the mutant strains. A lesser reduction in the binding affinity to secondary sites causes a displacement of Sir3 from the high affinity binding sites in the WT, to weaker secondary binding sites. These have binding affinities in the same order of magnitude as the compromised primary binding sites in some mutant strains. This allows the redistribution of Sir3 to secondary sites that are randomly distributed throughout the genome, fortuitously repressing genes near the new loci, including genes where null mutants were implicated in CLE. We therefore propose that the redistribution of heterochromatin domains are not mechanistically directly responsible for an extended lifespan, but that the incidental repression of genes that regulate lifespan cause the observed CLE.

## Methods

### Yeast strains and media

The histone mutant library, constructed by the Boeke group [23], and based on the parental strain JDY86 (*MATa, his3Δ200, leu2Δ0, K2Δ0, trp1Δ63, ura3Δ0, met15Δ0, can1::MFA1pr-HIS3, hht1-hhf1::NatMX4, hht2-hhf2:: [HHTS-HHFS]-URA3*, where [*HHTS-HHFS*] designates the mutated histone which is either H3 or H4) was purchased from Thermo Fischer Scientific. YPD yeast growth medium was prepared with 1% (w/v) yeast extract, 2% (w/v) peptone and 2% (w/v) glucose. Agar-YPD contained 2% (w/v) bacto-agar. All media were sterilized by autoclaving before use, and all chemicals were molecular biology grade.

### Culturing of the pooled library

The histone H3 and H4 mutant library was replica plated onto omnitray YPD-agar plates, and selected with 200 ng/ml nourseothricin (Sigma) antibiotic. Colonies were grown for 2–3 days at 30˚C. Slow-growing colonies were separately streaked out from the original stock, and grown for 2–3 days at 30˚C. The contents of all the plates, including the slow-growers (equivalent volume of a normal colony) were scraped off, and pooled in a 50 ml conical centrifuge tube containing YPD liquid media with 200 ng/ml nourseothricin. The pooled culture was diluted to a final concentration of $OD_{600} = 50$, 15% (v/v) glycerol was added, and stored as aliquots at -80˚C.

The pooled culture was inoculated into 100ml YPD liquid media to a final concentration of $OD_{600} \sim 0.003$. Cells were grown with rotary shaking (180 rpm) at 30˚C for approximately 10 generations (~15 h). The pooled culture was then further diluted to $OD_{600}$ of 0.06 in 250 ml YPD liquid media, and incubated for up to 55 d with continual shaking (180 rpm) at 30˚C. Media evaporation was minimized by using aluminum foil caps. Aliquots (10 ml) were recovered at the times indicated.

### Bar-code sequencing

Culture aliquots were washed twice with 5 ml water, the cells resuspended in 1 ml of 50 mM Tris-HCl (pH 7.5), and carefully overlaid onto a preformed Percoll gradient [24], and centrifuged at 400 g, 60 min, 20˚C in a GH-3.8 swinging bucket rotor (Beckman). Quiescent cells

were collected with a Pasteur pipette and washed once with 30 ml of 50 mM Tris-HCl pH 7.5 at 650 g, 5 min, 20˚C in a GH-3.8 swinging bucket rotor. Genomic DNA was isolated from 1 ml quiescent fractions (YeaStar kit, Zymo Research, Protocol I), and the DNA eluted with 60 μl of TE buffer (10 mM Tris-HCl pH 8.0, 0.1 mM EDTA), and stored at -80˚C. The "up" and "down" bar-code sequences were separately amplified in two reaction volumes containing 50 ng of genomic DNA, 0.3 μl of 5 U/μl FastStart Taq DNA polymerase (Roche), 0.4 μl of 100 μM "up" or "down" barcode primer pair, 0.5 μl of 10 mM dNTP, 12 μl of 25 mM $MgCl_2$, 3 μl of 10x PCR buffer without $MgCl_2$, and water to in a final volume of 30 μl. The primer sequences were ATGTCCACGAGGTCTCT, CCTCGACCTGCAGCGTA, CGGTGTCGGTCTC GTAG, and CCCAGCTCGAATTCATC for "up", forward and reverse, and "down", forward and reverse, respectively. The amplified DNA was purified [55], quantitated, single-end adapters with index primers (Illumina) ligated to each sample, and single-end sequenced on a HiSeq 2500 (ARC Biotechnology Platform, University of Pretoria).

The number of times that each barcode sequence was present in the FASTQ file was assessed for "up" and "down" barcodes with *get_seqs_from_fastq* for each time point, quantitated with *match_barcodes_to_strain*, and the average of the "up" and "down" value, adjusted for the amplified DNA signal, expressed as the $\log_2$ ratio relative to the level in the mixed starting population. The code is available at github.com/hpatterton. The data was deposited in the NCBI GEO archive (accession number GSE140160).

## Verification of chronological lifespan

To verify the results of the barcode analysis, 250 ml of YPD liquid media was inoculated to an $OD_{600}$ of 0.06 with starter cultures of selected mutant strains, and incubated for up to 50 d at 30˚C with shaking (180 rpm). Aliquotes (100 μl) of cells were recovered at sequential times, serially diluted onto YPD-agar plates, and the number of colony forming units (CFU/ml) determined after incubating for 2 d at 30˚C. Biological replicates were quantitated.

## RNA-seq

Total RNA was isolated from 10 ml cultures of yeast strains and purified with an RNeasy kit (Qiagen). The RNA was prepared for sequencing using the TruSeq Stranded Total RNA Sample Prep adaptor kit (Illumina). The samples were sequenced on a HiSeq 2500 instrument (Illumina) using the 100 nt paired-ends sequencing protocol. Biological replicates of each sample were independently prepared, sequenced and analysed.

The quality of the sequence reads was assessed with FastQC, and reads trimmed and filtered for a minimum Q score and length with Trim Galore. Reads were mapped to version 64 of the reference yeast genome (www.yeastgenome.org) with RNA STAR. The number of mapped reads per gene was counted with featureCounts, and differential expression quantitated with edgeR. The RNA-seq data was deposited in the NCBI GEO archive (accession number GSE141975).

## ChIP-seq

The ChIP experiments were performed on the bulk cells because the fraction of non-quiescent cells at the onset of stationary phase (day 6) is significantly less than the quiescent cells in the long living cells. Cultures (50 ml) were cross-linked in 1% (v/v) formaldehyde for 20 m, 25˚C, and the reaction quenched with glycine (125 mM), 5 m, 25˚C. The cells were washed once in 50 ml PBS (pH 7.4), and resuspended in lysis buffer (50 mM HEPES-KOH pH 7.4, 140 mM NaCl, 1mM EDTA, 1% (v/v) Triton-X-100 and Complete-mini EDTA free protease inhibitor cocktail [Roche]). Cells were lysed with 300 μl acid washed glass beads (425–600 μm) by

vigorous agitation for 30 s in a BeadRuptor2 followed by 30 s on ice, repeated 7 times. Sufficient lysis (90%) was confirmed by light microscopy. The lysate was subsequently sonicated at 4˚C to obtain an average DNA fragment length of 300 bp. Samples were centrifuged and the supernatant, containing sheared chromatin, recovered.

Chromatin was pre-cleared with 20 μl protein A/G magnetic beads (Invitrogen), at 4˚C for 1 hour. Pre-cleared chromatin was incubated with 10 μg of anti-Sir3 polyclonal IgG antibody (Genscript) or anti-Rap1 polyclonal IgG antibody (Abcam) for 30 min before 25 μl of the protein A/G magnetic beads was added. Samples were immuno-precipitated overnight at 4˚C with rotation. The supernatant was removed and the beads washed 3× in wash buffer (0.1% [w/v] SDS, 1% [v/v] TritonX-100, 20 mM EDTA, 20 mM Tris-HCl pH 8.0, 150 mM NaCl) followed by 1× wash in wash buffer 2 (0.1% [w/v] SDS, 1% [v/v] TritonX-100, 2 mM EDTA, 20 mM Tris-HCl pH8.0, 500 mM NaCl), and finally resuspended in elution buffer (1% [w/v] SDS, 100 mM $NaHCO_3$). The supernatant was incubated at 65˚C with 90 μg proteinase K, 4 h. Samples were subsequently incubated at 37˚C, 30 min after the addition of 35 μg RNase A. DNA was purified by phenol-chloroform extraction, precipitated in absolute ethanol, and stored at -80˚C.

The Rap1 samples were sequenced using a standard paired-ends protocol described by the manufacturer (Illumina) on a HiSeq250 instrument. For the Sir3 samples, a sequencing library was prepared with a NEXTflex DNA Sequencing kit (PerkinElmer) according to the manufacturer's protocol. End-repaired DNA fragments were ligated to IonCode Barcode Adapters (PerkinElmer), amplified, and sequenced to a median read length of 300 nt on an Ion S5 system. The read quality was assessed and the reads filtered as for the RNA-seq technique, the filtered reads mapped to version 64 of the *S. cerevisiae* reference yeast genome with Bowtie2, and peaks identified relative to a matched input sample with MACS2. The generated bedgraph files were converted to wiggle format, and the total signal in the immune-precipitated and input files adjusted to the identical sum. An identical normalisation was performed for different strains to allow cross comparison. The ChIP-seq analysis was performed on two biological replicates. The Rap1 and Sir3 ChIP-seq data were deposited in the NCBI GEO archive (accession numbers GSE141306 and GSE141317, respectively).

## Data analysis and statistics

Various C++ and Python 3 programs and scripts were developed to perform data conversions and analyses and are all available from http://www.github.com/hpatterton. Get_seqs_from_-fastq was used to count the number of occurrences of each histone mutant barcode in a FASTQ file. GO category enrichment was determined with GOrilla [56] and pathway analysis with KEGG Mapper [57]. Bedgraph files generated by MACS2 were converted to wiggle format and normalized with the convert_bedgraph_to_wiggle and normalize_wiggle programs.

## Supporting information

**S1 Fig. Interactions of residues implicated in CLE.** H3E50 makes hydrogen bonds to H4R39 (**a**). H3R128 and H3R131 make H-bonds to H3Y99 and H3D106 in α2, and H3R131 makes an H-bond to R99 in the Cα helix of H2A (**b**). H3K115 is oriented towards a phosphate group in the DNA sugar-phosphate backbone (**c**). H3I112 forms a hydrophobic pocket with H2AQ112 and H2AV114 (**d**). H3Q76 makes bridging H-bonds to the backbone at H3D81 in L1 and H3R72 in α1 (**e**). R83 of H3 is inserted into the minor groove at SHL ±2.5 (**f**). H3P66 at the N-terminus of α1 may contribute to the termination the α1 helix (**g**). H3Q55 is hydrogen bonded to the H2A C-terminal tail backbone at N110 as well as to R40 in α1 of H4 (**h**). H4R36 make salt bridge contacts with a phosphate group in the DNA sugar-phosphate backbone (**i**). H4R40

makes H-bonds to the H3 backbone αN and αN-α1 connecting coil (**j**). H4I50 is inserted between H3I119 and H3I124 (**k**). H4H75 in α2 of H4 make hydrophobic contact with H2BL77 and H2BT93, as well as an H-bond to H2BR89 (**l**). H4F100 makes numerous hydrophobic contacts with H3S87, H3A91, H4A83, H4V87 and H4L97 (**m**). H3F104 is inserted into a hydrophobic cluster formed by H3I119, H4L37 and H4V43 (**n**). Hydrophobic contacts between H3L60 and H3Q93 and between H3Q68 and H3V89 stabilizes the conformation of H3 α1 relative to H3 α2 (**o**).
(TIF)

**S2 Fig. Distribution of Rap1 in the genome during chronological aging.** The normalised binding of Rap 1 is shown at 16 h and 14 d for the left telomeric 15 kb of chromosome XII **(a)** and to the terminal 20 kb (tel) or remaining central region (core) of all 16 chromosomes normalised to the genome average **(b)**. The box plots represent the inter-quartile range (IQR), the notch represents the significance at p<0.05, the orange line shows the median, and the whiskers is shown at 1.5× the IQR. Outliers are shown as individual data points. There is no significant difference (p<0.05, *t*-test) between the corresponding data groups.
(TIF)

**S3 Fig. Genes that are covered by Sir3 in different combinations of the H4K16Q, H4H18A, H3E50A and WT strains.** The Venn diagram shows the number of genes associated with Sir3 peaks that are common to various combinations of the H4K16Q, H4H18A, H3E50A and WT strains.
(TIF)

**S1 Table. The viability of different histone mutant strain in stationary phase.** The ratio of the barcode associated with each different strain was normalised to that of the same barcode at time t0, and expressed as $\log_2$. The rows are sorted in descending order on the 44 d column.
(DOCX)

**S2 Table. Significance of transcription factor binding sites present in redistributed Sir3 peaks.** The number of transcription factor binding sites present within assigned Sir3 peaks was determined and the statistical significance of this number $p(k)$ calculated from a Poisson distribution, $e^{-} \cdot \frac{\lambda^k}{k!}$, where $k$ is the number of occurrences per unit length in a Sir3 peak, and $\lambda$ is equal to the number of genomic occurrences of the site per unit length.
(DOCX)

**S3 Table. Annotated genes that are associated with elevated genomic Sir3 levels in WT, H4K16Q, H4H18A and H3E50A yeast strains.**
(DOCX)

**S4 Table. GO biological function categories that show significant enrichment for genes that are induced (adjusted p<0.05) in the H4K16Q, H4H18A and H3E50A mutant strains compared to the WT strain.** The reported probability value is calculated as the probability mass function of a hypergeometric distribution using the values in the Enrichment column, and the q-value is the Benjamini–Hochberg corrected p-value.
(DOCX)

**S5 Table. KEGG pathways associated with genes that are significantly upregulated (adjusted p<0.05) in the H4K16Q, H4H18A and H3E50A mutant compared to the WT strain.** *S. cerevisiae* KEGG pathways were identified with KEGG Mapper[1], and are arranged by descending mapping frequency.
(DOCX)

**S6 Table. GO biological function categories that show significant enrichment for genes that are repressed (adjusted p<0.05) in the H4K16Q, H4H18A and H3E50A strains compared to the WT strain.** The reported probability value is calculated as the probability mass function of a hypergeometric distribution using the values in the Enrichment column, and the q-value is the Benjamini–Hochberg corrected p-value.
(DOCX)

**S7 Table. KEGG pathways associated with genes that are repressed (adjusted p<0.05) in the H4K16Q, H4H18A and H3E50A mutant compared to the WT strain.** *S. cerevisiae* KEGG pathways were identified with KEGG Mapper, and are arranged by descending mapping frequency.
(DOCX)

**S1 Note. Analysis of structure and interactions of residues implicated in chronological lifespan.**
(DOCX)

## Acknowledgments

We thank Junbiao Dai for providing reconstructed WT JD47 yeast strain, Dawie van Niekerk for advice on binding equilibria, and Riaan de Witt for critical reading of the manuscript.

## Author Contributions

**Conceptualization:** Hugh–George Patterton.

**Data curation:** Mzwanele Ngubo, Hugh–George Patterton.

**Formal analysis:** Mzwanele Ngubo, Jessica Laura Reid, Hugh–George Patterton.

**Funding acquisition:** Hugh–George Patterton.

**Investigation:** Mzwanele Ngubo, Jessica Laura Reid, Hugh–George Patterton.

**Methodology:** Mzwanele Ngubo.

**Project administration:** Hugh–George Patterton.

**Software:** Hugh–George Patterton.

**Supervision:** Hugh–George Patterton.

**Validation:** Mzwanele Ngubo.

**Writing – original draft:** Hugh–George Patterton.

**Writing – review & editing:** Mzwanele Ngubo, Jessica Laura Reid, Hugh–George Patterton.

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
