## [Decision Letter · Decision Letter 0]

2 Dec 2021

PONE-D-21-32183Distinct structural groups of histone H3 and H4 residues have divergent effects on chronological lifespan in Saccharomyces cerevisiaePLOS ONE

Dear Dr. Patterton,

Thank you for submitting your manuscript to PLOS ONE. After careful consideration, we feel that it has merit but does not fully meet PLOS ONE’s publication criteria as it currently stands. Therefore, we invite you to submit a revised version of the manuscript that addresses the points raised during the review process.

Reviewer 2 was more critical than Reviewer 1. However, the AE concurs with the issues raised by this Reviewer. In particular, the following revisions must be made:1. Distinguish between a change in lifespan vs altered kinetics of growth.2. Rule out a pseudo-diploid effect.3. Determine whether secondary Sir3 sites reported here are consistent with the literature.4. Compare exponential with stat phase for wild type and histone strains to check for reproducibility of data relative to previous studies.5. Show the transcriptome analysis as a figure in the main text6. Provide a more detailed Materials and Methods and more complete references to the literature.

Also address the additional issues raised by each of the reviewers.

We look forward to receiving your revised manuscript.

Kind regards,

Arthur J. Lustig, PhD

Academic Editor

PLOS ONE

Journal Requirements:

" ext-link-type="uri" xlink:type="simple">https://journals.plos.org/plosone/s/file?id=ba62/PLOSOne_formatting_sample_title_authors_affiliations.pdf"

This work was partially supported by the National Institutes of Health (Grant 

1U01HG007465, to HGP). The funding body did not contribute to the design of the study, 

collection, analysis, and interpretation of data, or to writing the manuscript. We thank 

Junbiao Dai for providing reconstructed WT JD47 yeast strain, Dawie van Niekerk for 

advice on binding equilibria, and Riaan de Witt for critical reading of the manuscript.

HGP

1U01HG007465

National Institutes of Health

https://www.nih.gov/

The funding body did not contribute to the design of the study, collection, analysis, and interpretation of data, or to writing the manuscript. 

Reviewers' comments:

Reviewer's Responses to Questions

**Comments to the Author**

1. Is the manuscript technically sound, and do the data support the conclusions?

Reviewer #1: Yes

Reviewer #2: Partly

2. Has the statistical analysis been performed appropriately and rigorously? 

Reviewer #1: Yes

Reviewer #2: I Don't Know

3. Have the authors made all data underlying the findings in their manuscript fully available?

Reviewer #1: Yes

Reviewer #2: Yes

4. Is the manuscript presented in an intelligible fashion and written in standard English?

Reviewer #1: Yes

Reviewer #2: Yes

5. Review Comments to the Author

Reviewer #1: This is a thoughtful and complete study analyzing the affect of histone mutations on cell aging and the related physiology and gene expression changes. It is a very well conceived and executed study and I learned a lot from reading it. I have only major comment.

1.These must have been a mistake in uploading the figures. Figure 2 is missing and Figure 4 is presented in its place. This is true in the merged document and the linked file.

Reviewer #2: In this manuscript, Patterton and colleagues have systematically tested the impact of histone point mutants on chronological lifespan (CLS). They identified 4 structural groups of mutants with distinct effects on lifespan. The paper focuses on the residues increasing lifespan. One group of residues showing chronological lifespan expansion (CLE) is involved in the interaction with the yeast heterochromatin factor Sir3. They performed ChIP experiments on representative mutants from which they conclude that mutants belonging to group 1 show a redistribution of Sir3 at secondary sites in the genome. Some of these new Sir3 binding sites are associated with genes previously reported to increase CLS when deleted. This could represent a direct cause to the observed phenotype. Surprisingly, the author performed transcriptome analysis in late log phase, which analysis is not shown in the main figures. Transcript levels of Sir3 bound genes should be shown as a main figure, especially the one that could be responsible for lifespan expansion.

In general the paper si not very well organized and some relevant references are missing (see below).

Specific comments:

1. The literature describing chromatin reorganization in quiescent cells is not cited (e.g. papers from the Tsukiyama and Taddei lab), ChIP against Sir3 in histone point mutants published by the Grunstein lab for exponential growing cells should be cited and compared with the results presented here.

2. The material and methods section is incomplete:

- The authors should explain how cells are grown over 50 days in flasks. How is the evaporation compensated for? Is the volume adjusted on a regular basis? If not, there must be some volume variation, thus affecting cell concentration and potentially biasing the lifespan assay. A detailed procedure is required here.

- It is not clear whether the ChIP experiments were performed on sorted Quiescent cells as for the bar-code sequencing (sorted on percoll gradient) or not.

3. Figure 2: Some strains show an increase of CFU during the first days, meaning that they did not reach saturation at day 2. This is a problem as we may not look at an increased lifespan but just a delay of the culture. Once they have reached a maximum, CFUs decrease faster than the wild type.

4. For the mutants showing redistribution of Sir3 on chromatin it would be important to rule out that the observed effect is not due to some pseudo-diploid effect that are observed when the HM loci are derepressed. This would require repeating the lifespan experiments for at least one mutant in a strain deleted for the relevant HM. The authors alluded to this potential caveat in the discussion but ruled it out by citing ref 48, which is about replicative lifespan and not chronological lifespan.

5. Sir3 secondary binding sites have been reported upon Sir3 overexpression (Radman-Livaja et al. 2011; Hocher et al. 2018; Ruault et al, 2021). Are the secondary sites reported here similar? What is the prediction from the model shown in Figure 7 ?

6.Also, it would have been nice to have the exponential culture conditions as a comparison with the stationary phase (6 days) for the ChIP experiments, at least for the WT strain and histone mutants for which these data sets are available in the literature (see point 1).

7. I found very surprising that the transcriptome analysis is not shown as a main figure. Transcript levels of the genes associated with new Sir3 binding sites, and previously described to impact lifespan when deleted (GDH1, AGP1, HTZ1 and SHR5 genes) should be shown as a main figure.

Minor comments:

Figure 1a: The trace of the log2 levels of the parental WT and median strain are not possible to see.

Line 183: maybe it is not the tail which is necessary but just the H3K4 (as stated lane 195-6).

Lane 227:

-It is not clear why the chip-seq was done at 6 days. It should have been compared to the exponential cultures. Are the Sir3 peaks identified later at 6 days already present in expo? What about the transcriptional status of those genes? When they are affected in late log phase, we would like to know what was their expression levels in expo and at day 6.

-Also: maybe repeat here that 6 days is stationary phase (not obvious for everybody).

Lane 233: Sir3 seems to be enriched in the Y’ element in the H4K16Q strain compared to other? any comments?

Figure 4: the representation of the statistical data is not clear. Should use other character to differentiate the groups or colors. It is difficult to read it as it is presented now. Since the figures are not heavy, I suggest to plot the count density from the core on a separated plot, in addition to the existing plot to have a better resolution..

Figure 5:

-The figure is not really in agreement with the statement that Rap1 levels are not changing in the H18 mutant ( it goes from 2 to 3.2 for Rap1, For Sir3, going from 1.2 to 1.35 was significant).

-This is a rather rough analysis. It would be important to show Rap1 profiles over chromosomal regions and to compare Rap1 peaks in the different conditions (expo phase and stationary phase).

Lane 298: it is worth mentioning that Abf1 and Rap1 are also found at HM silencers. However, there are many other transcription factors associated with these Sir3 peaks. Only Abf1 and Rap1 are mentioned but they are not the most significant (according to table 2).

Supplementary table 1 is not available.

Supplementary table 3 : whenever possible, gene’s name should be provided (like in sup2)

Lane 557: replace “performed” by “preformed”

6. PLOS authors have the option to publish the peer review history of their article (what does this mean?). If published, this will include your full peer review and any attached files.

Reviewer #1: No

Reviewer #2: No

---

## [Author Response · Author response to Decision Letter 0]

16 Jan 2022

PONE-D-21-32183

Distinct structural groups of histone H3 and H4 residues have divergent effects on chronological lifespan in Saccharomyces cerevisiae

PLOS ONE

Reviewer #1: This is a thoughtful and complete study analyzing the affect of histone mutations on cell aging and the related physiology and gene expression changes. It is a very well conceived and executed study and I learned a lot from reading it. I have only major comment.

1.These must have been a mistake in uploading the figures. Figure 2 is missing and Figure 4 is presented in its place. This is true in the merged document and the linked file.

We thank the reviewer for the comments and the correction, this has been rectified.

Reviewer #2: In this manuscript, Patterton and colleagues have systematically tested the impact of histone point mutants on chronological lifespan (CLS). They identified 4 structural groups of mutants with distinct effects on lifespan. The paper focuses on the residues increasing lifespan. One group of residues showing chronological lifespan expansion (CLE) is involved in the interaction with the yeast heterochromatin factor Sir3. They performed ChIP experiments on representative mutants from which they conclude that mutants belonging to group 1 show a redistribution of Sir3 at secondary sites in the genome. Some of these new Sir3 binding sites are associated with genes previously reported to increase CLS when deleted. This could represent a direct cause to the observed phenotype. Surprisingly, the author performed transcriptome analysis in late log phase, which analysis is not shown in the main figures. Transcript levels of Sir3 bound genes should be shown as a main figure, especially the one that could be responsible for lifespan expansion.

In general the paper si not very well organized and some relevant references are missing (see below).

Specific comments:

1. The literature describing chromatin reorganization in quiescent cells is not cited (e.g. papers from the Tsukiyama and Taddei lab), ChIP against Sir3 in histone point mutants published by the Grunstein lab for exponential growing cells should be cited and compared with the results presented here.

We thank the reviewer for the suggestions. Accordingly, we have cited the work from the Taddei and Tsukiyama lab in lines 110-111. We are currently unable to compare the Grunstein lab data with our results, the GEO data is in an old BAR format, and we have tried a number of utilities but could not find a script or program that could successfully convert the bar format to wig. We would appreciate it if the reviewer can suggest a script or program that he/she knows work.

2. The material and methods section is incomplete:

-The authors should explain how cells are grown over 50 days in flasks. How is the evaporation compensated for? Is the volume adjusted on a regular basis? If not, there must be some volume variation, thus affecting cell concentration and potentially biasing the lifespan assay. A detailed procedure is required here.

Evaporation of the media was minimized by using aluminum foil caps as previously described by Hu 2012, Cell Senescence; Barre 2020, Genome Res., shown in line 561-562. Volume adjustment by introducing fresh media would disrupt the steady state conditions and would encourage cells to re-enter the cycle, exiting G0 phase which is important in examining chronological aging.

-It is not clear whether the ChIP experiments were performed on sorted Quiescent cells same as for the bar-code sequencing (sorted on percoll gradient) or not.

The fraction of quiescent cells in stationary phase is an order of magnitude less than non-quiescent cells, because of this very big difference we have performed the ChIP experiments in bulk cells. We have addressed this in the new line 607-608. We have attached an image of the percoll gradient at exponential and stationary phase showing the difference as an illustration to the editor.

3. Figure 2: Some strains show an increase of CFU during the first days, meaning that they did not reach saturation at day 2. This is a problem as we may not look at an increased lifespan but just a delay of the culture. Once they have reached a maximum, CFUs decrease faster than the wild type.

The initial increase of cells during the first days can be attributed to the mother cells undergoing one more asymmetric cell division after exponential phase (Werner-Washburne 1993, Microbiol Rev; Herman 2002 Curr Opin Microbiol; Miles 2020, Yeast Extract). The initial increase in chronologically aged yeast is observed by many other groups in the field (Picazo 2015, Plos ONE; Wierman 2017, MCB; Ogawa 2016, PNAS). Since the cells are gown at equivalent initial concentration the initial increase is inferred to fitness of the strain in growth limiting stationary phase culture conditions. We use the definition of chronological lifespan as described in the field by Kennedy, Longo, Werner-Washburne, Guarente, Sinclair, and others that define chronological aging as the length of time a population of yeast cells remains viable in a non-dividing state following nutrient deprivation. 

4. For the mutants showing redistribution of Sir3 on chromatin it would be important to rule out that the observed effect is not due to some pseudo-diploid effect that are observed when the HM loci are derepressed. This would require repeating the lifespan experiments for at least one mutant in a strain deleted for the relevant HM. The authors alluded to this potential caveat in the discussion but ruled it out by citing ref 48, which is about replicative lifespan and not chronological lifespan.

A thorough search of the literature did not reveal any study that investigated the effect of an E or I element deletion, and expected associated derepression of HMLalpha (or HMRa) on chronological lifespan. We did identify a single study where extension of replicative lifespan was associated with increased HM repression (Kim et al 2017, MBoc). Thus, although we could not specifically identify a study addressing chronological lifespan, all published data that relate lifespan and HM silencing suggest a direct relationship, furthermore, the loss of HM silencing has been showed to be an effect of aging (Kennedy et al.,1997, Sinclair and Guarente, 1997, Smeal et al., 1996). Also, deletion of MATalpha2 causes derepression of HMLalpha. However, this strain is not associated with extension of chronological lifespan in phenotypic survey studies (Laschober et al 2010, Aging Cell; Campos et al 2018, Aging Cell), strongly suggesting that derepression of HM is not causative of chronological lifespan extension. We do not suspect that mating sterility, a phenotype of HM derepression could be a cause of longevity, whereas strong evidence suggest that it is an effect.

5. Sir3 secondary binding sites have been reported upon Sir3 overexpression (Radman-Livaja et al. 2011; Hocher et al. 2018; Ruault et al, 2021). Are the secondary sites reported here similar? What is the prediction from the model shown in Figure 7? 

There is minimal overlap of the Sir3 secondary binding sites in the histone mutants and the secondary binding sites induced by Sir3 overexpression (Hocher et al. 2018; Ruault et al, 2021), suggesting that the recruitment of Sir3 to the alternative sites that cause lifespan extend does not resemble the same recruitment patterns as in cells with increased concentrations of Sir3. Out of 18 genes in the Sir3 euchromatin sites in the overexpressed Sir3 background strains, only 3 genes overlap with the histone mutants’ secondary site genes, and none in the extended silent domain genes. Therefore, our proposed model demonstrates systematic redistribution of Sir3 to regions that are key for CLE compared to ectopic Sir3 expression. Furthermore, overexpression of Sir3 is not associated with chronological lifespan extension (Butcher et al, 2006 Nature Chemical Biology; Yoshikawa et al 2011, Yeast).

6. Also, it would have been nice to have the exponential culture conditions as a comparison with the stationary phase (6 days) for the ChIP experiments, at least for the WT strain and histone mutants for which these data sets are available in the literature (see point 1).

Please see response to point1

7. I found very surprising that the transcriptome analysis is not shown as a main figure. Transcript levels of the genes associated with new Sir3 binding sites, and previously described to impact lifespan when deleted (GDH1, AGP1, HTZ1 and SHR5 genes) should be shown as a main figure.

We have shown normalized transcript levels of genes linked with new Sir3 binding sites, and implicated in lifespan extension in new Fig6.

Minor comments:

Figure 1a: The trace of the log2 levels of the parental WT and median strain are not possible to see.

We have shown the median strain in the new Fig1a and Fig1e.

Line 183: maybe it is not the tail which is necessary but just the H3K4 (as stated lane 195-6).

Other residues outside of the H3K1-4 region also showed decreased lifespan, see Figure 1i, suggesting that the tail is important for normal lifespan, not only a single H3K4 residue.

Line 227:

-It is not clear why the chip-seq was done at 6 days. It should have been compared to the exponential cultures. Are the Sir3 peaks identified later at 6 days already present in expo? What about the transcriptional status of those genes? When they are affected in late log phase, we would like to know what was their expression levels in expo and at day 6.

Stationary phase day 6 is day 0 as far as lifespan is concerned, thus, we were interested in the state of chromatin and distribution of Sir3 at the point where we started measuring chronological lifespan.

As we have noted in line 352-356, in yeast, there is a well-documented transcriptional shutdown in stationary phase. We appreciate that transcription or repression of certain genes may underpin CLE, however, we were interested in the transcriptome profile that preceded stationary phase, and that may prime cells for later lifespan extension in stationary phase.

-Also: maybe repeat here that 6 days is stationary phase (not obvious for everybody).

We have reiterated this accordingly in the new line 233.

Line 233: Sir3 seems to be enriched in the Y’ element in the H4K16Q strain compared to other? any comments?

Our quantification of the total signal over the entire Y’ element shows comparable Sir3 enrichment in the H4K16Q strain to the other strains as indicated in Fig3c. The slight difference in enrichment could be strain specific and is non-significant.

Figure 4: the representation of the statistical data is not clear. Should use other character to differentiate the groups or colors. It is difficult to read it as it is presented now. Since the figures are not heavy, I suggest to plot the count density from the core on a separated plot, in addition to the existing plot to have a better resolution.

We have presented the data in a standard box plot with multiple t-test comparisons. The p-value symbols are clearly illustrated and explained in the figure legend.

Figure 5:

-The figure is not really in agreement with the statement that Rap1 levels are not changing in the H18 mutant (it goes from 2 to 3.2 for Rap1, For Sir3, going from 1.2 to 1.35 was significant).

The reviewer should kindly remember that the statistical t-test analysis does not look at only the means of the samples but it also takes into account the standard deviations and the number of the set size, which all factor in the calculation of statistical significance (p-value) in the changes observed.

-This is a rather rough analysis. It would be important to show Rap1 profiles over chromosomal regions and to compare Rap1 peaks in the different conditions (expo phase and stationary phase).

Please see Supplementary Fig. 2a where we showed Rap1 peaks for the different conditions over 15kb chromosome XII chromosomal region. 

Line 298: it is worth mentioning that Abf1 and Rap1 are also found at HM silencers. However, there are many other transcription factors associated with these Sir3 peaks. Only Abf1 and Rap1 are mentioned but they are not the most significant (according to table 2).

For the purpose of this study, we focused on Abf1 and Rap1 because of their known biological importance in directly interacting with Sir3.

Supplementary table 1 is not available.

The Supplementary table 1 has been included.

Supplementary table 3 : whenever possible, gene’s name should be provided (like in sup2)

Gene names have been provided where possible. See Supplementary table 3.

Line 557: replace “performed” by “preformed”

We thank the reviewer for catching typo, it has been corrected.

---

## [Decision Letter · Decision Letter 1]

22 Feb 2022

PONE-D-21-32183R1Distinct structural groups of histone H3 and H4 residues have divergent effects on chronological lifespan in Saccharomyces cerevisiaePLOS ONE

Dear Dr. Patterton,

Thank you for submitting your manuscript to PLOS ONE. After careful consideration, we feel that it has merit but does not fully meet PLOS ONE’s publication criteria as it currently stands. Therefore, we invite you to submit a revised version of the manuscript that addresses the points raised during the review process.

 The authors failed to respond to most of the comments  of Reviewer 2 and there was no point-by-point  rebuttal to this reviewers. The authors must provide this information for further consideration as described in the revision preparation rules. (Please examine original AE letter of revision).Of additional concern are data that do not include appropriate statistical evaluation as noted by Reviewer 2. Additional terminology was be explained as pointed out by Reviewer 2.Please submit your revised manuscript by Apr 08 2022 11:59PM. If you will need more time than this to complete your revisions, please reply to this message or contact the journal office at plosone@plos.org. Please include the following items when submitting your revised manuscript:A rebuttal letter that responds to each point raised by the academic editor and reviewer(s). You should upload this letter as a separate file labeled 'Response to Reviewers'.A marked-up copy of your manuscript that highlights changes made to the original version. You should upload this as a separate file labeled 'Revised Manuscript with Track Changes'.An unmarked version of your revised paper without tracked changes. You should upload this as a separate file labeled 'Manuscript'.

If applicable, we recommend that you deposit your laboratory protocols in protocols.io to enhance the reproducibility of your results. Protocols.io assigns your protocol its own identifier (DOI) so that it can be cited independently in the future. For instructions see: https://journals.plos.org/plosone/s/submission-guidelines#loc-laboratory-protocols. Additionally, PLOS ONE offers an option for publishing peer-reviewed Lab Protocol articles, which describe protocols hosted on protocols.io. Read more information on sharing protocols at https://plos.org/protocols?utm_medium=editorial-emailutm_source=authorlettersutm_campaign=protocols.

We look forward to receiving your revised manuscript.

Kind regards,

Arthur J. Lustig, PhD

Academic Editor

PLOS ONE

Journal Requirements:

Additional Editor Comments (if provided):

It is absolutely critical that the comments of all reviewers are addressed. Reviewer 2's critique had no response from the authors. This is not acceptable.

Reviewers' comments:

Reviewer's Responses to Questions

**Comments to the Author**

1. If the authors have adequately addressed your comments raised in a previous round of review and you feel that this manuscript is now acceptable for publication, you may indicate that here to bypass the “Comments to the Author” section, enter your conflict of interest statement in the “Confidential to Editor” section, and submit your "Accept" recommendation.

Reviewer #2: (No Response)

2. Is the manuscript technically sound, and do the data support the conclusions?

Reviewer #2: Partly

3. Has the statistical analysis been performed appropriately and rigorously? 

Reviewer #2: No

4. Have the authors made all data underlying the findings in their manuscript fully available?

Reviewer #2: Yes

5. Is the manuscript presented in an intelligible fashion and written in standard English?

Reviewer #2: Yes

6. Review Comments to the Author

Reviewer #2: This new version of the manuscript is very similar to the previous one and most of my comments were not considered. This is clearly illustrated by the track-change version of the manuscript in which only few changes can be seen. Yet, I have some comments on these changes:

p6: "In addition, quiescence has been shown to be essential in determining the chromatin dynamics involved in longevity."

This sentence (added in response to my first comment) does not fil well in this section, it is difficult to understand why this information is provided at this stage and not in the introduction.

p29: The ChIP experiments were performed on the bulk cells because the fraction of quiescent cells in stationary phase is significantly less than the non-quiescent cells."

Is this correct? If true, the ChIP is thus performed (mainly)) on non-quiescent cells? Is this proportion the same for all strains?

Figure 2 has been corrected in response to reviewer’s comment. Indeed, this figure was missing in the first version. It would be important to show the variability of these growth curves. Although the main text and the figure legend specify that growth curves were done in biological replicate, there is no error bars in the plot.

Figure 6 has been completed to answer one of my comments (point 7). Panel b and c have been added to show the normalized transcript levels of genes linked with new Sir3 binding sites and implicated in lifespan extension.

First, here again there are no error bars. Second, from this figure one can clearly see that the GDH1 gene is not repressed in the H4K16 mutant, contrary to what is written l332-333

“The GDH1 glutamate dehydrogenase gene is repressed in the H4K16 mutant strain and is associated with a CLE phenotype in a null strain”; the MAM3 and YIL055C that are repressed in the H4K16 mutant (according to Figure 6), but these genes don’t induce CLE when deleted (as stated l314 and L316 respectively). Therefore, there is no ‘simple, causative link between Sir3 redistribution and lifespan expansion” for the H4K16 mutant (in contradiction with l338).

Regarding the H4H18 mutant, only SHR5 is bound by Sir3, repressed, and associated with CLE.

Comments on the non-answered comments:

Among the 7 main points raised in my review only 2 have been completely addressed (point 2 and 7)

Point 1 and 6: Sir3 ChIP seq data are readily accessible for WT cells in exponential phase and it would be important to compare these with Sir3 ChIP on stationary phase cells. As for the data from the Grunstein lab on histone mutants, it is always possible to re-analyze the raw data using the software and parameters described in the original paper, and export the files in the appropriate format.

Point 3: I don’t understand how the authors can say that they “define chronological

aging as the length of time a population of yeast cells remains viable in a non-dividing state

following nutrient deprivation », while their interpret “The initial increase of cells during the first days […] to the mother cells undergoing one more asymmetric cell division after exponential phase.

Point 4: In their lengthy response to this point, the authors continue to confuse CLS and RLS and do not really answer the main question, which is whether there is a pseudo-diploid effect.

Point 5: I found interesting that “There is minimal overlap of the Sir3 secondary binding sites in the histone mutants and the secondary binding sites induced by Sir3 overexpression”. I think this should be mentioned in the manuscript and whether it was predicted by their model or not.

7. PLOS authors have the option to publish the peer review history of their article (what does this mean?). If published, this will include your full peer review and any attached files.

Reviewer #2: No

---

## [Author Response · Author response to Decision Letter 1]

19 Apr 2022

Reviewers' comments:

Reviewer's Responses to Questions

Reviewer #2: This new version of the manuscript is very similar to the previous one and most of my comments were not considered. This is clearly illustrated by the track-change version of the manuscript in which only few changes can be seen. Yet, I have some comments on these changes:

p6: "In addition, quiescence has been shown to be essential in determining the chromatin dynamics involved in longevity."

This sentence (added in response to my first comment) does not fil well in this section, it is difficult to understand why this information is provided at this stage and not in the introduction.

We have moved the sentence to the introduction. See line 74-76.

p29: The ChIP experiments were performed on the bulk cells because the fraction of quiescent cells in stationary phase is significantly less than the non-quiescent cells."

Is this correct? If true, the ChIP is thus performed (mainly)) on non-quiescent cells? Is this proportion the same for all strains?

We thank the reviewer for seeking this clarification, indeed, we have made an error in the mentioned statement. For the long living cells, the fraction of non-quiescent cells at the onset of stationary phase in day 6 (where we performed our Sir3 ChIP experiments) is significantly less than the quiescent cells, however, at day 55 as the quiescent cells age and accumulate damage the non-quiescent cells increase. Accordingly, the ChIP experiments were performed mainly on quiescent cells at day6 in stationary phase. This proportion difference is the same for five of the tested top10 strains. We have attached an image (Appendix 1) with representative strains for illustration at day2, day6 and day55 for the mutant strains, and day2, day7 and day34 for WT. 

Figure 2 has been corrected in response to reviewer’s comment. Indeed, this figure was missing in the first version. It would be important to show the variability of these growth curves. Although the main text and the figure legend specify that growth curves were done in biological replicate, there is no error bars in the plot.

New Figure 2 with error bars showing the standard deviation has been included.

Figure 6 has been completed to answer one of my comments (point 7). Panel b and c have been added to show the normalized transcript levels of genes linked with new Sir3 binding sites and implicated in lifespan extension.

First, here again there are no error bars. Second, from this figure one can clearly see that the GDH1 gene is not repressed in the H4K16 mutant, contrary to what is written l332-333

“The GDH1 glutamate dehydrogenase gene is repressed in the H4K16 mutant strain and is associated with a CLE phenotype in a null strain”; the MAM3 and YIL055C that are repressed in the H4K16 mutant (according to Figure 6), but these genes don’t induce CLE when deleted (as stated l314 and L316 respectively). Therefore, there is no ‘simple, causative link between Sir3 redistribution and lifespan expansion” for the H4K16 mutant (in contradiction with l338).

Regarding the H4H18 mutant, only SHR5 is bound by Sir3, repressed, and associated with CLE.

We thank the reviewer for correctly pointing this out. The bar graphs in (Figure 6b, c) were made from the FPKM normalization of the count data which is not suitable for showing transcriptomic differences across samples. We have now showed a new (Figure 6c, d) with the bar graph of log2 fold change from differential expression analysis using the DESeq2 package which is suitable for between-samples comparison. New Figure 6c shows significant repression of GDH1 gene in the H4K16 mutant. The MAM3 and YIL055C genes are not repressed in both H4K16 and H4H18 mutant strains (Figure 6c, d). Additionally, AGP1 and HTZ1 genes are repressed in the H4H18 mutant strain consistent with the original text. 

Regarding the error bars, DESeq2 employs the Wald statistical test where the estimated standard error of a log2 fold change is tested. The Wald statistical test use the biological replicates we have performed. Moreover, we attach an image (Appendix 2) illustrating the dispersion of variance and the density distribution of the counts, suggesting that the datasets are appropriate to detect small degrees of variability between the levels of expression in different sets.

Comments on the non-answered comments:

Among the 7 main points raised in my review only 2 have been completely addressed (point 2 and 7)

Point 1 and 6: Sir3 ChIP seq data are readily accessible for WT cells in exponential phase and it would be important to compare these with Sir3 ChIP on stationary phase cells. As for the data from the Grunstein lab on histone mutants, it is always possible to re-analyze the raw data using the software and parameters described in the original paper, and export the files in the appropriate format.

We thank the reviewer for the suggestion to compare our data with publicly available exponential phase Sir3 ChIP-seq data. Indeed, as we expected the Sir3 occupancy at HM loci is enhanced in the exponential and stationary phase WT compared to the H4K16 and H4H18 mutant strains, accordingly, upstream of SND1 there is reduction in Sir3 occupancy in the exponential and stationary phase WT compared to the H4K16 and H4H18 mutant strains. See new (Figure 3b, d) and Figure 6b. 

Concerning the Grunstein lab data on histone mutants, as we have mentioned in our previous response, the Affymetrix Tiling Array Software raw data outputs binary files (.bar and .cel) which we have tried different scripts to export to bigwig files, with no success. We even reached out to the first author who told us they never exported the bar files to bigwig files.

Point 3: I don’t understand how the authors can say that they “define chronological

aging as the length of time a population of yeast cells remains viable in a non-dividing state following nutrient deprivation », while their interpret “The initial increase of cells during the first days […] to the mother cells undergoing one more asymmetric cell division after exponential phase.

The reviewer expressed a concern that some of the strains increased after day2 and therefore did not reach saturation, which suggests that the authors are looking at delayed cultures. To reiterate our response, we state that the initial increase of cells observed in our cultures is characteristic of post-diauxic growth where mother cells in the culture undergo one more cell division or slow growth into daughter cells that will become quiescent cells (Werner-Washburne 1993, Microbiol Rev; Herman 2002 Curr Opin Microbiol; Miles 2020, Yeast Extract). The survival of the initially increased cells farther into stationary phase is due to the fitness of the long living strains in culture conditions that are not favourable for growth, and not delayed growth as suggested by the reviewer. In stationary phase, glucose is exhausted and the nutrients are limited, deterring growth of unfit strains as seen in the WT and the short living strains, unless the strains are fit to withstand these culture conditions as seen in long living mutant strains.

Point 4: In their lengthy response to this point, the authors continue to confuse CLS and RLS and do not really answer the main question, which is whether there is a pseudo-diploid effect.

We apologize to the reviewer for not being clear in our response. We will try to unpack our response again here. The pseudo-diploid effect occurs as a result of derepression of the MATa and MATalpha genes at the HM loci in haploid cells, which has been suggested to cause mating type defect or mating sterility (Dörfel 2017, Yeast). Our argument is that if the pseudo-diploid effect causes sterility, an effect of aging, in RLS we don't suspect that it would cause longevity in CLS. More importantly, the overexpression of MATalpha1 gene in a haploid MATa strain was associated with short lived strains in CLS (Pogoda 2021, Biogerentology), strongly suggesting that derepression of HM loci, or the pseudo-diploid effect is not causative of chronological lifespan extension. We have replaced the RLS reference with the CLS study. See line 470-471.

Point 5: I found interesting that “There is minimal overlap of the Sir3 secondary binding sites in the histone mutants and the secondary binding sites induced by Sir3 overexpression”. I think this should be mentioned in the manuscript and whether it was predicted by their model or not.

We are happy to mention this in our discussion. See line 494-499.

---

## [Decision Letter · Decision Letter 2]

9 May 2022

Distinct structural groups of histone H3 and H4 residues have divergent effects on chronological lifespan in *Saccharomyces cerevisiae*

PONE-D-21-32183R2

Dear Dr. Patterton,

We’re pleased to inform you that your manuscript has been judged scientifically suitable for publication and will be formally accepted for publication once it meets all outstanding technical requirements.

Kind regards,

Arthur J. Lustig, PhD

Academic Editor

PLOS ONE

Additional Editor Comments (optional):

Reviewers' comments:

Reviewer's Responses to Questions

**Comments to the Author**

1. If the authors have adequately addressed your comments raised in a previous round of review and you feel that this manuscript is now acceptable for publication, you may indicate that here to bypass the “Comments to the Author” section, enter your conflict of interest statement in the “Confidential to Editor” section, and submit your "Accept" recommendation.

Reviewer #2: All comments have been addressed

2. Is the manuscript technically sound, and do the data support the conclusions?

Reviewer #2: Yes

3. Has the statistical analysis been performed appropriately and rigorously? 

Reviewer #2: Yes

4. Have the authors made all data underlying the findings in their manuscript fully available?

Reviewer #2: Yes

5. Is the manuscript presented in an intelligible fashion and written in standard English?

Reviewer #2: Yes

6. Review Comments to the Author

Reviewer #2: In this new version of their manuscript, the authors have responded to most of my comments in a convincing manner.

7. PLOS authors have the option to publish the peer review history of their article (what does this mean?). If published, this will include your full peer review and any attached files.

Reviewer #2: No

---

## [Editor Report · Acceptance letter]

13 May 2022

PONE-D-21-32183R2 

Distinct structural groups of histone H3 and H4 residues have divergent effects on chronological lifespan in *Saccharomyces cerevisiae*

Dear Dr. Patterton:

I'm pleased to inform you that your manuscript has been deemed suitable for publication in PLOS ONE. Congratulations! Your manuscript is now with our production department. 

Kind regards, 

on behalf of

Dr. Arthur J. Lustig 

Academic Editor

PLOS ONE